# A novel notion of barycenter for probability distributions based on optimal weak mass transport

**Elsa Cazelles**
IRIT, Université de Toulouse, CNRS
`elsa.cazelles@irit.fr`

**Felipe Tobar**
IDIA & CMM, Universidad de Chile
`ftobar@dim.uchile.cl`

**Joaquin Fontbona**
CMM, Universidad de Chile
`fontbona@dim.uchile.cl`

## Abstract

We introduce weak barycenters of a family of probability distributions, based on the recently developed notion of optimal weak transport of mass [26], [9]. We provide a theoretical analysis of this object and discuss its interpretation in the light of convex ordering between probability measures. In particular, we show that, rather than averaging the input distributions in a geometric way (as the Wasserstein barycenter based on classic optimal transport does) weak barycenters extract common geometric information shared by all the input distributions, encoded as a latent random variable that underlies all of them. We also provide an iterative algorithm to compute a weak barycenter for a finite family of input distributions, and a stochastic algorithm that computes them for arbitrary populations of laws. The latter approach is particularly well suited for the *streaming setting*, i.e., when distributions are observed sequentially. The notion of weak barycenter and our approaches to compute it are illustrated on synthetic examples, validated on 2D real-world data and compared to standard Wasserstein barycenters.

## 1 Introduction

Optimal transport (OT) [41] has had a tremendous impact in the machine learning (ML) community recently, as it provides meaningful and implementable distances between probability distributions [34], thus advancing many aspects in the field, see e.g. [7, 43, 33]. The space of probability measures on $\mathbb{R}^d$ with finite second moment can be *metrised* with the Wasserstein-2 distance, the computation of which amounts to finding a transport plan that minimises the quadratic average cost of transporting mass from a source probability measure onto a target one. In this context, a natural method for averaging a finite family of probability measures is to compute their Fréchet mean, with respect to the Wasserstein-2 distance, which corresponds to the *Wasserstein barycenter* introduced in [1].

The goal of the present work is to explore theoretical features and potential applications to ML of barycenters of probability measures analogously defined in terms of *optimal weak transport* (OWT, see [26]) or more precisely quadratic barycentric transport costs. In a nutshell, for a source measure $\mu$ and a target measure $\nu$, the OWT problem aims to transport mass so that the conditional spatial mean of target support points $y$, given their source support points $x$, is close to $x$ in average. This amounts to finding an intermediate measure $\eta$, possibly *more concentrated* than $\nu$ in the sense of convex ordering of probability measures, which is *close* to $\mu$ with respect to the Wasserstein-2 distance.

The main motivation of our work is to investigate the effect and meaning of combining a family of probability measures using OWT instead of OT. To that end, we will define the *weak barycenter* of

35th Conference on Neural Information Processing Systems (NeurIPS 2021).

this family through an optimisation problem, and discuss some of its properties. Importantly, we will see that, rather than averaging the input distributions in a metric sense, solving a weak barycenter problem corresponds to finding probability measures that encode geometric or shape information *shared across* all of them. In fact, the weak barycenter problem will be interpreted as finding a latent random variable common to all the input distributions. Implications of this latent variable interpretation, in terms of robustness to outliers, will also be drawn in our work.

A second motivation for our work is to develop and implement computational methods for weak barycenters, capitalising on the fact that the optimal weak coupling between *any* pair of distributions, with finite second moments, is always realised by a unique optimal *map*. This property is in sharp contrast to standard OT, where the absolute continuity with respect to the Lebesgue measure of the source or target measure is typically needed to grant the existence and uniqueness of a map—the so-called Monge map— realising the optimal coupling between them. This map is often required in different ways to compute Wasserstein barycenters (see [5], [44] or [35]).

Similarly to the Wasserstein barycenter problem, we will develop a fixed-point formulation of the weak barycenter problem, based on OWT plans. This allows us to, following [5, 44], construct an iterative procedure to compute a weak barycenter for a finite family of distributions and analyse its convergence properties. We will also define and study the so-called *weak population barycenters*, that deal with a population of probability measures distributed according to a given law $\mathbb{Q}$ supported on the Wasserstein-2 space, as in [30] for the OT case. Extending ideas from [10], we will then propose an iterative stochastic algorithm for online computation of the weak population barycenter, from a *stream* of probability measures sampled from $\mathbb{Q}$. We will then provide numerical simulations using this proposed method, in order to illustrate the geometric meaning of the weak barycenter, and we will compare it with related objects obtained with standard OT or its entropy-regularised counterpart.

**Organisation of the paper.** Sec. 2 documents the background on OWT and the assumptions underlying our work. Sec. 3 analyses the weak barycenter problem, interprets it in the light of convex ordering and a latent variable model and addresses the case of an infinite population of distributions. Sec. 4 introduces two algorithms for computing the weak barycenter in the finite or population settings. Sec. 5 and 6 present the experimental setting and validation of our proposal respectively. Lastly, Sec. 7 discusses our findings and future research questions. The Appendix contains all the proofs, additional details or our simulations and the code of our experiments.

## 2 Background : optimal weak transport and Wasserstein barycenter

The optimal transport (OT) problem [42] aims to find the lowest cost to transfer the mass from one probability measure onto another. Therefore, OT is a natural way to compare two probability distributions in terms of their geometric information. In particular, the Wasserstein-$p$ distance $W_p$, associated with the Euclidean cost in $\mathbb{R}^d$, metrises the space $\mathcal{P}_p(\mathbb{R}^d)$ of probability measures on $\mathbb{R}^d$ with finite $p$-moment. Precisely, for $\mu, \nu \in \mathcal{P}_p(\mathbb{R}^d)$,

$$W_p(\mu, \nu) = \left( \min_{\pi \in \Pi(\mu,\nu)} \int_{\mathbb{R}^d \times \mathbb{R}^d} \|x - y\|^p \mathrm{d}\pi(x, y) \right)^{1/p}, \tag{1}$$

where $\pi$ is a *transport plan* between $\mu$ and $\nu$, that is, an element of the set $\Pi(\mu, \nu)$ of probability measures on the product space $\mathbb{R}^d \times \mathbb{R}^d$ with marginals $\mu$ and $\nu$. For $p = 2$ and $\mu$ absolutely continuous (*a.c.*), the unique optimal plan is concentrated on the graph of a measurable map called *M*onge map such that $\nu = T\#\mu$, see eq. (14) in Appendix A.1.

**Optimal weak transport.** We consider here the optimal weak transport (OWT) problem introduced in [26] and in particular the special case of barycentric transport costs. The OWT problem is then defined for $\mu, \nu \in \mathcal{P}_2(\mathbb{R}^d)$ by

$$V(\mu|\nu) = \inf_{\pi \in \Pi(\mu,\nu)} \int_{\mathbb{R}^d} \|x - \int_{\mathbb{R}^d} y \mathrm{d}\pi_x(y)\|^2 \mathrm{d}\mu(x), \tag{2}$$

where $\pi_x$ is the *disintegration* of the transport plan $\pi$ with respect to the first marginal $\mu$, *i.e.* $\pi(\mathrm{d}x\mathrm{d}y) = \pi_x(\mathrm{d}y)\mu(\mathrm{d}x)$. As our work strongly leans on OWT theory, we recall in Appendix A.2, Th. 6, that $V$ is continuous with respect to the Wasserstein metric [9]. Additionally, the two following results from [8] (stated for our specific setting) lay the ground for our proposed weak barycenters.

**Theorem 1** ([8], Theorem 1.2). *The problem* (2) *admits a unique minimiser.*

This first result strongly differs from the classical OT setting, for which the uniqueness of an optimal transport plan is not guaranteed for arbitrary measures. The optimisation problem in Eq. (2) can also be reformulated thanks to the Brenier-Strassen theorem [25], [8], through the notion of convex ordering. We denote by $\eta \leq_c \nu$ the *convex order of measures*, meaning that $\int \phi d\eta \leq \int \phi d\nu$ for any convex function $\phi$ that is nonnegative or integrable with respect to $\eta + \nu$. By Strassen's theorem [40], two distributions are in convex order if and only if there exists a martingale coupling between them. The following theorem is a generalisation of the result originally proved in [25], Th. 1.2.

**Theorem 2** ([8], Theorem 1.4). *Let $\mu \in \mathcal{P}_2(\mathbb{R}^d)$ and $\nu \in \mathcal{P}_1(\mathbb{R}^d)$. There exists a unique $\eta^* \leq_c \nu$ such that*

$$W_2^2(\mu, \eta^*) = \inf_{\eta \leq_c \nu} W_2^2(\mu, \eta) = V(\mu|\nu). \tag{3}$$

*Moreover, there exists a convex function $\psi : \mathbb{R}^d \to \mathbb{R}$ of class $C^1$ with $\nabla\psi$ being $1$-Lipschitz, such that $\nabla\psi\#\mu = \eta^*$. Finally, the optimal coupling $\pi^{\mu,\nu} \in \Pi(\mu,\nu)$ verifies $\int y d\pi_x^{\mu,\nu}(y) = \nabla\psi(x)$ $\mu$-a.s.*

The measurable map, or barycentric projection, $S_\mu^\nu(x) := \int_{\mathbb{R}^d} y d\pi_x^{\mu,\nu}(y)$ associated to the plan $\pi^{\mu,\nu}$ achieving the minimum in Eq. (2) is consequently uniquely defined and will be called *optimal barycentric projection*. From this notation, we can write the OWT cost in terms of an OT cost according to $V(\mu|\nu) = W_2^2(\mu, S_\mu^\nu\#\mu)$. We emphasise that $S_\mu^\nu$ is directly related to the optimisation problem (2), whereas applied works such as [38, 36] make use of a barycentric projection constructed from a transport plan solving an OT problem between $\mu$ and $\nu$ (often regularised) as a substitute for the Monge map, which may not exist (more details on $S_\mu^\nu$ are displayed in Appendix A.3).

Last, let us note that OWT is somehow also related to the martingale OT problem developed in the stochastic finance community [13, 2, 27], which puts the focus on the optimal transfer of mass between distributions assumed to be in convex order themselves.

**Wasserstein barycenter.** The classical Wasserstein barycenter problem for a set of probability measures $\nu_1, \ldots, \nu_n \in \mathcal{P}_2(\mathbb{R}^d)$ with weights $\lambda_1, \ldots, \lambda_n$ in the simplex (i.e. $\lambda_i \geq 0$ and $\sum_{i=1}^n \lambda_i = 1$) is defined [1] by

$$\arg\min_{\mu \in \mathcal{P}_2(\mathbb{R}^d)} \sum_{i=1}^n \lambda_i W_2^2(\mu, \nu_i). \tag{4}$$

The Wasserstein barycenter has been extensively studied both theoretically and numerically [30, 44, 5, 15]. Regarding the numerical part, [39] focuses on the computation of Wasserstein barycenters for a fixed number of measures and a stream of observations per measure; additionally, [32] proposed an entropy-regularised alternative via stochastic optimisation for computing the Wasserstein barycenter of *a.c.* distributions only from observations. Constrained by their assumption of *a.c.*, [44] computes the Wasserstein barycenter by smoothing the observed empirical distributions. Furthermore, [21] compares the complexity of both the *sample* Wasserstein barycenter and a stochastic approximation to estimate a population barycenter (discrete measures and entropic regularisation). Finally, the authors of [3] recently proposed an algorithm to compute the barycenters in polynomial time.

## 3 Optimal weak transport barycenters and latent variable interpretation

### 3.1 Definition and basic properties

In a similar fashion, based on the weak transport cost in Eq. (2), we propose the following variant:

**Definition 1.** *The set of weak barycenters of a finite family of measures $\{\nu_i\}_{i=1,\ldots,n} \in \mathcal{P}_2(\mathbb{R}^d)$ with weights $\{\lambda_i\}_{i=1,\ldots,n}$ in the simplex is defined as*

$$\arg\min_{\mu \in \mathcal{P}_2(\mathbb{R}^d)} \sum_{i=1}^n \lambda_i V(\mu|\nu_i). \tag{5}$$

Thus, a weak barycenter averages, with respect to the Wasserstein metric, an optimally chosen set of probability measures $\{\eta_1, \ldots, \eta_n\}$ which are *more concentrated* than the corresponding $\nu_i$, in the sense that $\eta_i \leq_c \nu_i$ for each $1 \leq i \leq n$. The existence of a solution is established as follows:

**Proposition 1.** *The weak barycenter problem in Eq. (5) admits a minimiser $\mu \in \mathcal{P}_2(\mathbb{R}^d)$.*

See Sec. B of the Appendix for the proof of the above Proposition (which relies on Prokhorov's theorem) and all the proofs for this Section. Uniqueness is in general not granted: we next show that the set of solutions is indeed an interval, with respect to the partial order of convex ordering of probability measures.

In the following, we denote by $X$ and $Y_i$ random variables with respective laws $\mu$ and $\nu_i$, for $1 \leq i \leq n$, and $\delta_a$ the Dirac measure supported on $a \in \mathbb{R}^d$.

**Lemma 1.** *If $\mu$ is a weak barycenter of $\{\nu_i\}_{i=1...,n}$ and $\mu' \leq_c \mu$, then $\mu'$ also is a weak barycenter. In particular, the Dirac measure supported on $\mathbb{E}_\mu(X)$ is always a weak barycenter. Moreover, a Dirac distribution $\delta_{\bar{\omega}}$ is a weak barycenter if and only if $\bar{\omega} = \sum_{i=1}^n \lambda_i \mathbb{E}_{\nu_i}(Y_i)$.*

A consequence of the above lemma is that for any weak barycenter $\mu$,

$$\mathbb{E}_\mu(X) = \sum_{i=1}^n \lambda_i \mathbb{E}_{\nu_i}(Y_i),\tag{6}$$

and the value of the weak barycenter problem is given by

$$\inf_{\mu \in \mathcal{P}_2(\mathbb{R}^d)} \sum_{i=1}^n \lambda_i V(\mu|\nu_i) = \sum_{i=1}^n \lambda_i \|\mathbb{E}(Y_i)\|^2 - \|\sum_{i=1}^n \lambda_i \mathbb{E}(Y_i)\|^2.\tag{7}$$

We can also derive the following characterisation on the set of weak barycenters:

**Proposition 2.** *A measure $\mu \in \mathcal{P}(\mathbb{R}^d)$ is a weak barycenter of $\{\nu_i\}_{i=1...,n}$ if and only if its mean satisfies (6) and $\hat{\mu} \leq_c \hat{\nu}_i$ holds for all $1 \leq i \leq n$, where $\hat{\nu}$ denotes the centered version of a law $\nu$.*

For instance, in the case of one dimensional Gaussian distributions $\nu_i = \mathcal{N}(m, \sigma_i^2)$, the set of weak barycenters includes $\{\mu = \mathcal{N}(m, \sigma^2) \mid 0 \leq \sigma^2 \leq \min_{1 \leq i \leq n} \sigma_i^2\}$.

A natural question is whether a "maximal" weak barycenter exists, in the sense of convex ordering (up to translation by the mean). For $d = 1$, the answer is affirmative. When the means $\mathbb{E}(Y_i)$ are equal, this follows from the complete lattice property of the set of probability measures with respect to the convex ordering (see [29]); the general case can then be reduced to the latter using Proposition 2. For $d \geq 2$, this property is in general not true and the answer depends on the family $\{\nu_i\}_{i=1...,n}$.

In the particular case of *a.c.* input measures, we can bound the distance between the Wasserstein and weak barycenters by the variances of the distributions $(\nu_i)_{1 \leq i \leq n}$. The barycenters are then closer the more concentrated each $\nu_i$ is.

**Lemma 2.** *Let $\nu_1, \ldots, \nu_n \in \mathcal{P}_2(\mathbb{R}^d)$ be a.c., at least one of them with bounded density. Let $\bar{\mu}$ and $\tilde{\mu}$ respectively denote the weak and the Wasserstein barycenters. Then*

$$W_2^2(\bar{\mu}, \tilde{\mu}) \leq 2 \sum_{i=1}^n \lambda_i \left( \mathbb{E}\|Y_i\|^2 - \|\mathbb{E}Y_i\|^2 \right).$$

## 3.2 Weak barycenters as latent variables

The weak barycenter encodes common geometric information present in all the input measures considered, therefore, it can be intuitively and rigorously interpreted as being the distribution of a latent variable underlying the realisations of random variables of laws $\nu_i$ for all $1 \leq i \leq n$.

**Theorem 3.** *Let $\mu$ be a weak barycenter of $\{\nu_i\}_{i=1...,n}$. Then, for each $1 \leq i \leq n$, a random variable $Y_i \sim \nu_i$ can be realised as*

$$Y_i = X + (\mathbb{E}Y_i - \mathbb{E}X) + \bar{Y}_i,$$

*where $X \sim \mu$ and $\bar{Y}_i = Y_i - \mathbb{E}(Y_i|X)$ is centered conditionally on $X$. Moreover, one has $S_\mu^\nu(X) = X + (\mathbb{E}Y_i - \mathbb{E}X)$ for all $i = 1, \ldots, n$. Finally, we have $\mathbb{E}(Y_i - \mathbb{E}Y_i|X - \mathbb{E}X) = X - \mathbb{E}X$ or, equivalently, $\hat{\mu} \leq_c \hat{\nu}_i$, with $\hat{\mu}$ and $\hat{\nu}_i$ the laws of $X - \mathbb{E}X$ and $Y_i - \mathbb{E}Y_i$ respectively.*

That is to say, each $Y_i \sim \nu_i$ can be realised by sampling a random variable $X$ common to all $i = 1, \ldots, n$ and distributed according to the weak barycenter $\mu$, translating that value by $\mathbb{E}Y_i - \mathbb{E}X$ and adding a *cluster-specific* component $\bar{Y}_i$ or idiosyncratic noise, centered conditionally on $X$.

**Remark 1.** *The observations of each class (i.e. input measure) can be interpreted as outliers with respect to the (translated) law of the weak barycenter, which are statistically different and are thus left aside of its support. This way, the weak barycenter is robust to outliers, as it tends to discard them, by construction. Furthermore, this "robustness" property results in the stability of weak barycenter upon perturbation of a class with larger noise (or more scattered, outlying values). More precisely, if a class is corrupted in such a way that their observations result in a stochastically larger distribution than the original one, a weak barycenter computed in terms of the original (stochastically smaller) class will still be a weak barycenter in the new corrupted setting. An intuitive and simple way to illustrate this point follows by considering a weak barycenter $\mu$ of a one dimensional and centered family of input distributions $\{\nu_i\}_{i=1,\ldots,n}$. By Proposition 2, $\mu$ must verify $\mu \leq_c \nu_i$ for all $i = 1, \ldots, n$. In particular, from Theorem 3.A.1. in [37], we have that $\int_x^\infty \mathbb{P}(X > u)du \leq \int_x^\infty \mathbb{P}(Y_i > u)du$ for all $x \in \mathbb{R}$, where $X \sim \mu$ and $Y_i \sim \nu_i$. Therefore, $\mu$ is likely to avoid outliers. Another supportive intuition in terms of robustness is that a maximal weak barycenter would be one that includes the most possible points of all classes (or distributions) in its support (all this, after re-centering) and leaves out only "outliers". A non-maximal weak barycenter is then more conservative, meaning that it counts on fewer points and leaves out more possible outliers.*

### 3.3 Extension for the population barycenter

The population Wasserstein barycenter introduced in [30] and [4] extends the definition of Wasserstein barycenter for an infinite number of measures. This formulation is particularly relevant for the construction of an iterative algorithm to compute the barycenter for the streaming case, that is, when the measures are received *online*. The proofs are reported in Section C of the Appendix.

Let us consider a probability measure $\mathbb{Q} \in \mathcal{P}_2(\mathcal{P}_2(\mathbb{R}^d))$, meaning that $\mathbb{Q}$ is supported on a set of measures with finite moments of order 2, such that for some (and thus all) $\mu \in \mathcal{P}_2(\mathbb{R}^d)$, we have that $\int_{\mathcal{P}_2(\mathbb{R}^d)} W_2^2(\mu, \nu)d\mathbb{Q}(\nu) < \infty$.

**Definition 2.** *We define the set of weak population barycenters of a distribution $\mathbb{Q} \in \mathcal{P}_2(\mathcal{P}_2(\mathbb{R}^d))$ as*

$$\operatorname*{arg\,min}_{\mu \in \mathcal{P}_2(\mathbb{R}^d)} \int_{\mathcal{P}_2(\mathbb{R}^d)} V(\mu|\nu)d\mathbb{Q}(\nu). \tag{8}$$

The following lemma guarantees that the map $(x, \nu) \mapsto S_\mu^\nu(x)$ appearing in Eq. (8) through $V(\mu|\nu) = \int \|x - S_\mu^\nu(x)\|^2 d\mu(x)$ is well defined.

**Lemma 3.** *The function $(\mu, \nu) \in (\mathcal{P}_2(\mathbb{R}^d))^2 \mapsto \pi^{\mu,\nu} \in \mathcal{P}_2(\mathbb{R}^{2d})$ mapping $(\mu, \nu)$ to the unique optimal plan $\pi^{\mu,\nu}$ realising $V(\mu|\nu)$ in Eq. (2) is continuous. As a consequence, for each $\mu \in \mathcal{P}_2(\mathbb{R}^d)$ the function $(x, \nu) \in \mathbb{R}^d \times \mathcal{P}_2(\mathbb{R}^d) \mapsto S_\mu^\nu(x)$ is measurable.*

Using similar arguments as those of Proposition 1 and the fact that any probability measure can be approximated by a sequence of probability measures with finite support, the following proposition confirms that the weak population barycenter problem is also well defined.

**Proposition 3.** *The minimisation problem in Eq. (8) admits a solution.*

## 4 Algorithms via fixed-point representations

### 4.1 Weak barycenter

For the Wasserstein barycenter problem in Eq. (4), the authors in [1] proved that if at least one of the measures $\nu_1, \ldots, \nu_n$ is *a.c.*, the Wasserstein barycenter is unique. Furthermore, if all the $\nu_i$'s are *a.c.*, and at least one of them has a bounded density, then the unique Wasserstein barycenter is also *a.c.* and verifies a fixed-point equation. This last property has been thoroughly studied by [5] and [44] and leveraged to compute an approximation of the barycenter via an iterative algorithm based on Monge maps, whose existence and uniqueness are guaranteed by the *a.c.* of the measures involved.

Akin to the fixed-point methodology in the classical Wasserstein scenario, we define an iterative procedure based on the barycentric projection computed in the optimal weak transport problem in Eq. (2), that is valid for arbitrary distributions. Therefore, we consider the following iterative rule for

probability measures $\nu_1, \dots, \nu_n \in \mathcal{P}_2(\mathbb{R}^d)$:

$$\mu_{k+1} = G(\mu_k), \text{ with } G(\mu) = \left( \sum_{i=1}^{n} \lambda_i S_\mu^{\nu_i} \right) \#\mu, \tag{9}$$

where for each $i = 1, \dots, n$, the optimal barycentric projection is given by $S_\mu^{\nu_i} : x \mapsto \int y \mathrm{d}\pi_x^{\mu,\nu_i}(y)$, for $\pi^{\mu,\nu_i} \in \Pi(\mu, \nu_i)$ achieving the minimum in the OWT problem in Eq. (2). The proposed iterative procedure is presented in Algorithm 1.

A fundamental difference between the fixed-point computation of the Wasserstein barycenter [5] and a weak barycenter is that the optimal Monge map $T_\mu^\nu$ in the OT problem verifies $T_\mu^\nu \#\mu = \nu$, whereas the pushforward measure $S_\mu^\nu \#\mu$ in the OWT setting still depends on $\mu$. We will then prove that the iterative algorithm in Eq. (9), based on the maps $S_\mu^{\nu_i}$, admits converging subsequences. A convenient result is the continuity of the functional $G$ in Eq. (9), which can be proven using Arzela-Ascoli theorem on a set of barycentric projections as well as the Skorohod's representation theorem.

**Theorem 4.** *The function $\mu \mapsto G(\mu)$ defined in Eq. (9) is $W_2$-continuous from $\mathcal{P}_2(\mathbb{R}^d)$ to $\mathcal{P}_2(\mathbb{R}^d)$.*

Using an approach similar to [5] for the Wasserstein barycenter, we can state the following results for the proposed fixed-point procedure.

**Proposition 4.** *If $\mu$ is a weak-barycenter, that is a solution of problem* (5)*, then $G(\mu) = \mu$ i.e. $x = \sum_{i=1}^{n} \lambda_i S_\mu^{\nu_i}(x), \mu(x)$-a.s.*

The inverse implication of Proposition 4 is not necessarily true, that is, some fixed points may not be weak barycenters. However, a Dirac delta $\delta_\omega, \omega \in \mathbb{R}^d$, that meets the fixed-point condition $\delta_\omega = G(\delta_\omega)$, is a weak barycenter (see Lemma 1).

**Proposition 5.** *Let $(\mu_k)_k$ be the sequence defined by the iterative procedure $\mu_{k+1} = G(\mu_k)$ and starting from $\mu_0 \in \mathcal{P}_2(\mathbb{R}^d)$. Then $(\mu_k)_k$ is tight and every converging subsequence must converge to a fixed point of $G$.*

We observe that these results also hold for the classical Wasserstein barycenter of *a.c.* measures $\{\nu_i\}_{i=1\dots,n}$ such that at least one of them has a bounded density. Moreover, the inverse implication, namely if $\mu$ is a fixed-point then it is a barycenter, is not straightforward even in the Wasserstein barycenter case, for which one considers the fixed-point equation given by $\mu = (\sum_{i=1}^{n} \lambda_i T_\mu^{\nu_i}) \#\mu$, with $T_\mu^{\nu_i}$ the Monge map verifying $\nu_i = T_\mu^{\nu_i} \#\mu$. Indeed, [1] prove that if $\mu$ checks $x = \sum_{i=1}^{n} \lambda_i T_\mu^{\nu_i}(x)$ for every $x \in \mathbb{R}^d$, not only $\mu$-almost everywhere, then $\mu$ is a Wasserstein barycenter. Also, [44, Theorem 2] provide additional conditions for this to be true by essentially invoking more smoothness on the distributions $\{\nu_i\}_{i=1\dots,n}$. Additionally, they only conjecture that under the same assumptions, the fixed-point is unique. Our method, however, includes arbitrary probability measures. Therefore, we do not expect to obtain similar results as in the Wasserstein barycenter case, for which smoothness is required.

## 4.2 Weak population barycenter

Based on [10], we construct a stochastic iterative algorithm for computing the weak population barycenter in Eq. (8). We clarify that [10] is constrained to probability measures $\mathbb{Q}$ supported on distributions that are *a.c.*, whereas in our setting these distributions only need to belong to $\mathcal{P}_2(\mathbb{R}^d)$. Let us notice that our algorithms can be interpreted as geodesic gradient descent as in [10] and [17], however, OWT is not a metric and its potential geodesic structure is so far unknown. Therefore, the proposed algorithm only aims to mimic Riemannian gradient descent. Our fixed-point result for the weak population barycenter problem is stated in the following Lemma:

**Lemma 4.** *If $\mu$ is a weak population barycenter of $\mathbb{Q}$, then $x = \int S_\mu^\nu(x) d\mathbb{Q}(\nu), \mu(x)$-a.s.*

As in the finite case, the inverse implication is difficult to obtain. In particular, this has not been proven for the classical population Wasserstein barycenter in [10], where it boils down to prove the uniqueness of an absolutely continuous fixed point of $\mu \mapsto (\int T_\mu^\nu d\mathbb{Q}(\nu)) \#\mu$, where $T_\mu^\nu$ is the Monge map between $\mu$ and $\nu$. As explained in [10], the uniqueness of such fixed points has also been studied under some strong assumptions in [15] by considering parametric classes of random probability measures with compact support. This result is expected to be true by again invoking more smoothness on the distributions at hand. As our method focuses (in particular) on discrete probability measures,

the conditions under which the inverse implication holds are beyond the scope of our work. However, from the experimental results in Section 6, we believe our method presents practical advantages.

We next present an iterative scheme converging towards a distribution $\mu$ verifying the fixed-point equation in Lemma 4. This scheme is illustrated below in Algorithm 2. To prove its convergence, we will need a technical assumption on $\mathbb{Q}$:

(A)    There exists $\epsilon > 0$ and $R > 0$ such that $\mathbb{Q}$ gives full measure to the set
$$K_{\mathbb{Q}} := \{\mu \in \mathcal{P}_2(\mathbb{R}^d) : \int |x|^{2+\epsilon} \mathrm{d}\mu(x) \leq R\}.$$

**Definition 3.** *Let* $\mu_0 \in K_{\mathbb{Q}}, \nu^k \overset{i.i.d.}{\sim} \mathbb{Q}$ *and* $\gamma_k > 0$*. We define the following iterative procedure:*

$$\mu_{k+1} = \left[(1 - \gamma_k)\mathrm{id} + \gamma_k S_{\mu_k}^{\nu^k}\right] \#\mu_k, \ k \geq 0, \tag{10}$$

*where* $S_{\mu_k}^{\nu^k}$ *is the optimal barycentric projection between* $\mu_k$ *and* $\nu^k$ *and* id *is the identity operator.*

The following standard conditions on the steps $\gamma_k$ will also be assumed:

$$\sum_{k=1}^{\infty} \gamma_k^2 < \infty \qquad \text{and} \qquad \sum_{k=1}^{\infty} \gamma_k = \infty, \tag{11}$$

**Theorem 5.** *Assume Conditions in Eq.* (11)*, (A) and moreover that every measure verifying the fixed-point equation* $x = \int S_\mu^\nu(x) d\mathbb{Q}(\nu), \mu(x)$*-a.s. is a weak barycenter. Then the sequence* $(\mu_k)_k$ *in Eq.* (10) *is a.s. relatively compact w.r.t.* $W_2$ *and every limit point is a weak barycenter.*

The proof, provided in the supplementary material, is inspired by the standard Wasserstein barycenter case studied in [10], [35]. Assumption (A) grants that the sequence in Eq. (10) remains in some compact set, and can be replaced by more general conditions (see Remark 2 in Appendix).

---

**Algorithm 1:** Weak barycenter

**Input:** distributions $\nu_1, \ldots, \nu_n$, # steps $K$;
initialisation: $\mu_0 = \nu_1$;
**for** $k = 0, 1, \ldots, K$ **do**
    **for** $i = 1, 2, \ldots, n$ **do**
        Solve the OWT problem between $\mu_k$
        and $\nu_i$ to obtain $\pi^{\mu_k, \nu_i}$;
        $S_i = \int y \mathrm{d}\pi_x^{\mu_k, \nu_i}(y)$
    **end**
    $\mu_{k+1} = (\sum_{i=1}^n \lambda_i S_i) \#\mu_k$
**end**

---

**Algorithm 2:** Weak population barycenter

**Input:** number of steps $K$;
initialise distribution $\mu_0 \sim \mathbb{Q}$;
**for** $k = 0, 1, \ldots, K$ **do**
    Sample $\nu^k \sim \mathbb{Q}$;
    Update $\gamma_k$;
    Solve the OWT problem to obtain $\pi^{\mu_k, \nu^k}$
    $S_k = \int y \mathrm{d}\pi_x^{\mu_k, \nu^k}(y)$;
    $\mu_{k+1} = \left[(1 - \gamma_k)\mathrm{id} + \gamma_k S_k\right] \#\mu_k$;
**end**

---

## 5   Computational aspects

**Setting and computation of OWTs.** Both Algorithms 1 and 2 require the computation of the optimal barycentric projection associated to the OWT problem in Eq. (2). For two discrete measures $\mu = \sum_{i=1}^r a_i \delta_{x_i}$ and $\nu = \sum_{j=1}^m b_j \delta_{y_j}$, this boils down to solving the following quadratic programming problem

$$\min_{\pi \in \mathbb{R}^{r \times m}} \left\{ \sum_{i=1}^r a_i \left\| x_i - \left(\frac{\pi \mathbf{y}}{\mathbf{a}}\right)_i \right\|^2, \pi_{ij} \geq 0, \ \pi \mathbb{1} = a, \ \pi^T \mathbb{1} = b \right\}, \tag{12}$$

which can be solved using a solver such as **cvxpy**. We also propose to solve the OWT problem in Eq. (12) with a proximal algorithm. The optimal barycentric projection is then constructed as $\frac{\pi \mathbf{y}}{\mathbf{a}}$. The details and examples are presented in Appendix E.1.

**Comparison setting.** In the next section, we compare our proposed computation for weak barycenters in Definition 2 (Algorithm 2) to the classic Wasserstein barycenter in particular for a stream of a.c. measures. Namely, we will run Algorithm 2 by, following [19, 36], replacing optimal barycentric

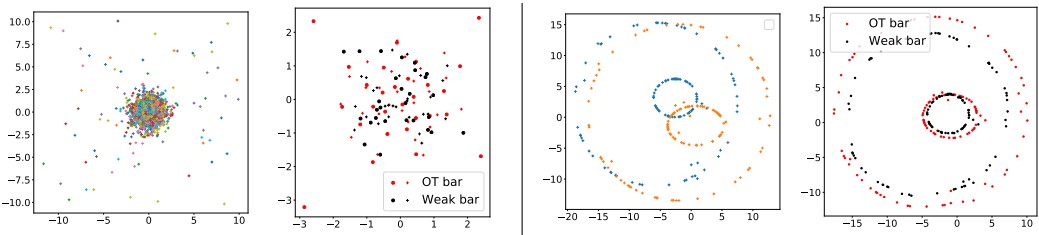

Figure 1: (left) Empirical Gaussian distributions and their OWT (black) and OT (red) barycenters for Gaussian observations (crosses) and corrupted observations (dots). (right) Empirical distributions supported on two ellipses and their OWT (black) and OT (red) barycenters.

projections by the barycentric projections associated either to i) an optimal plan in the Kantorovich problem (1), or ii) the optimal Sinkhorn plan in the entropy regularised OT problem [18] given by

$$\arg\min_{\pi \in \Pi(\mu,\nu)} \int \|x - y\|^2 \mathrm{d}\pi(x,y) + \varepsilon KL(\pi|\mu \otimes \nu), \tag{13}$$

where KL denotes the Kullback-Leibler divergence. The associated barycenters will be referred to as *OT barycenter* and *OT Sinkhorn barycenter* respectively. The optimal plans for OT and regularised OT problem were computed using *POT toolbox* [23]. Notice that what we call OT barycenter (resp. OT Sinkhorn barycenter) is not solving a Wasserstein barycenter problem (resp. a regularised Wasserstein barycenter problem). Therefore, our method for barycentric computation differs from previous ones in the literature (see Section 2) in that it i) can process a *stream* of an unknown number of measures, ii) does not require the measures to be a.c., and iii) does not appeal to additional regularisation of the measures or the Wasserstein metric.

## 6 Experimental results

This section is devoted to the empirical validation of our proposal on both synthetic and real-world data. We first focused on Algorithm 2 since multiple algorithms to compute a Wasserstein barycenter for a fixed number of distributions are already available [19, 39]. We present two robustness to outliers experiments, then we validate our OWT barycenter on synthetic dataset and real-world ones. The overall conclusion of our experiments is that the weak barycenter is more likely to maintain the common (or shared) geometric features of the measures involved, as expected from Theorem 3. Additional experiments are presented in Appendix E.2, including the comparison of the energy for the computed weak barycenter in Algorithm 1 against the approximated optimal energy (using Eq.(7) and the plug-in estimator).

### 6.1 Robustness to outliers

OT's sensitivity to outliers is a well-known problem that can be addressed *e.*g. with unbalanced OT [11]. We observed that OWT also allows to deal with outliers, which is coherent with the latent variable interpretation (see Remark 1). We illustrate this with two experiments. In Fig. 1 (left), we consider 50 sets of $20 - 30$ observations from different 2D Gaussian measures, where each observation may be corrupted by random translations (Bernoulli $p = 0.05$) thus producing outliers. We show the resulting barycenters (dots), and barycenters without outliers (crosses) for Wasserstein barycenter (red) and weak barycenter (black), which shows robustness to outliers. In Fig. 1 (right), we consider two distributions supported on pair-of-ellipses, and 120 observations per distribution. Again, each observation may be corrupted by random translations (Bernoulli $p = 0.05$). The weak barycenter (black) shows a better preservation of the shapes than the Wasserstein barycenter (red), in particular, the red dots are more often located outside the ellipses.

### 6.2 Synthetic distributions

We implemented the proposed sequential computation of weak barycenters (Algorithm 2) on two examples of synthetic distributions: Gaussians and spirals. In each case, we sampled $r$ observations from a random distribution at each step, and considered $K$ steps (and thus $K$ measures for each case).

**2D Gaussians** ($r = 100$ & $K = 15$). We considered distributions $\mathcal{N}(m, I)$, with $m$ uniformly distributed on $(-3, 3) \times (-5, -5)$ and $I$ the identity matrix. Fig. 2 (left) shows the empirical distributions together with the OWT and OT barycenters, the weak barycenter being the less spread out as expected. The three remaining plots illustrate the behaviour of the barycenters constructed as stated in Sec. 5. For a small regularisation parameter $\varepsilon$ in Eq. (13), the OT and OT Sinkhorn barycenters are similar, however, as $\varepsilon$ increases the OT Sinkhorn (OTS) barycenter becomes closer to the weak barycenter and thus even more concentrated, meaning that its samples tend to be closer to each other. Critically, for a very large $\varepsilon$, as the entropy tends to spread the mass in the regularised optimal plan, the associated barycentric projection will roughly move the mass to the spatial mean of the target distribution's support.

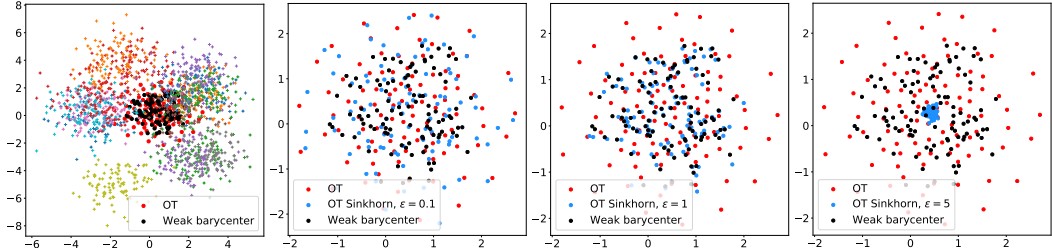

Figure 2: (left) Empirical Gaussian distributions and their OWT (black) and OT (red) barycenters computed with Algorithm 2. Illustration of the weak (black), OT (red) and OT Sinkhorn (blue) barycenters for different values of $\varepsilon = 0.1, 1, 5$.

**Spiral distributions.** ($r \in (200, 225)$ & $K = 10$). In this experiment, we considered distributions supported on a spiral— see Fig. 3 (left), with random ratio in $(0, 3)$. The OT and OWT barycenters are presented in Fig. 3 (right). Again, the weak barycenter seems to better preserve the shape of the spiral than the OT barycenter.

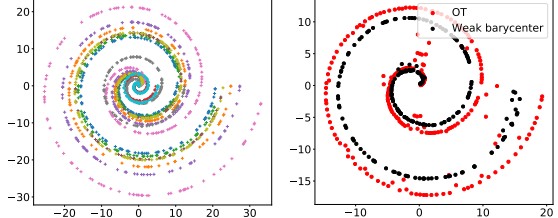

Figure 3: (left) Distributions supported on spiral. (right) OWT (black) and OT (red) barycenters computed with Algorithm 2.

### 6.3 Real-world dataset

**MNIST dataset.** We considered the well-known MNIST dataset [31] of grayscale images of handwritten digits. The images, of size $28 \times 28$ pixels, can be normalised and thus be interpreted as discrete probability measures supported on a two-dimensional grid of size $28 \times 28$. We computed the barycenters with 30 steps of Algorithm 1 between two digits "8", that are noisy versions of the same digit with the aim to produce a more stable barycenter. To produce noisy data, we randomly (Bernoulli $p = 0.1$) move pixels of the prototype digit displayed in Fig. 4 (left). Fig. 4 (right) shows the barycenters using the OWT, OT, and entropic-OT (for $\varepsilon = 1$). This example illustrates how OWT reduces dispersion, so that weak barycenter provides the best uniformly spread results among the barycenters considered, with the two loops of the "8" well shaped.

**Cytometry dataset.** In biotechnology, *flow cytometry* is measured through intracellular markers of single cells in a biological sample with the objective of recognising common features across patients. However, these measurements are often disrupted by acquisition, rather than biological artefacts [28], thus hindering the identification of common features. To address this challenge, we compute the weak barycenter for the forward-scattered light (FSC) and side-scattered light (SSC) cell's markers (using the flowStats package of Bioconductor [24]). We considered $K = 15$ patients and a variable number of cells per patient between 88 and 2185. Fig. 5 shows the 15 distributions (left) and the computed barycenters (right), thus confirming the ability of the weak barycenter to resolve the alignment of the dataset, while maintaining the expected diamond-shape. Moreover, the advantage of our proposed streaming procedure is fully exploited in this setting, since data from one or several patient can arrive

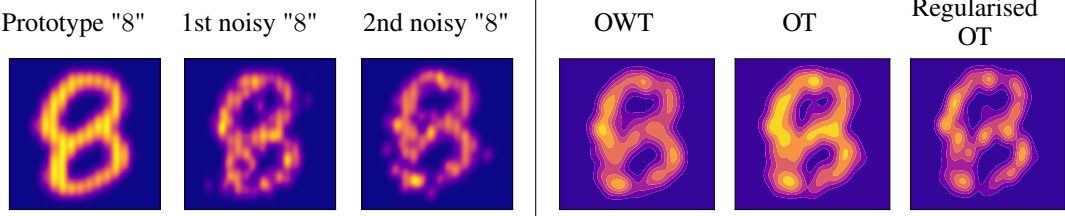

| Prototype "8" | 1st noisy "8" | 2nd noisy "8" | OWT | OT | Regularised OT |

Figure 4: Digit "8" (MNIST). From left to right: Prototype "8", first and second noisy versions of the prototype by randomly (Bernoulli $p = 0.1$) moving pixels, three barycenters constructed with Algorithm 1 associted to the OWT plan, an OT plan and the entropy regularised OT plan for $\varepsilon = 1$.

sequentially. Though this setting has been addressed with the Wasserstein barycenter in [14], also in Fig. 5, such method required a fixed grid to compute the barycenter unlike our method, thus revealing the computational simplicity of the weak barycenter.

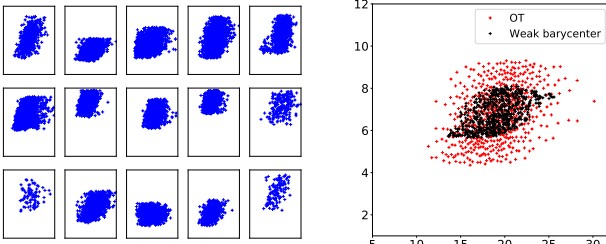

Figure 5: (left) Cytometry dataset for $n = 15$ patients and FSC vs. SSC cell's marker. (right) The weak-barycenter (black) computed with Algorithm 2 and the OT barycenter (red). The data are represented with the same axis as the figure of barycenters.

# 7    Discussion

We have introduced the weak barycenter, which extracts common geometric information of probability measures on $\mathbb{R}^d$ based on optimal weak transport, and showed that it can be interpreted as a latent variable model. From the fixed-point formulation defined in terms of optimal weak transport maps, irrespective of the regularity assumptions on the measures involved, we developed practical computation via an iterative algorithm with guaranteed convergence. In particular, the proposed algorithms do not require a common grid on the sample space, when processing either observed data or samples from distributions. We have also proposed weak barycenters of a possibly infinite population of measures and developed a stochastic procedure for computing it in the streaming data regime where distributions are processes into the weak barycenter as they arrive. This has critical implications for continual-learning methods in the ML community.

Additional studies will focus on deepen the latent variable interpretation of weak barycenters, and its relationship to the aggregate information represented by the Wasserstein barycenter. Also, we identify two relevant theoretical aspects for further research: i) to exhibit general conditions on the family of input measures (or on the law of the population) for the existence of weak barycenters that are not Dirac masses; and ii) to provide conditions on those input measures for a "maximal" weak barycenter (in terms of convex ordering) to exist when $d \geq 2$, among all the solutions of the weak barycenter problem (and, if possible, a way of constructing it by regularisation most probably). The statistical behaviour of the weak barycenter can also be investigated, in particular when constructed from large empirical random samples of given distributions. Lastly, the weak population barycenter could also be used to construct a predictive posterior in the context of Bayesian learning, as was done for Wasserstein barycenters in [35].

**Acknowledgments.** We thank Julio Backhoff-Veraguas for his valuable insight during the writing of this paper. This work was funded by ANID grants: AFB170001 & ACE210010 (CMM), FB0008 (AC3E), Fondecyt-Postdoctorado #3190926 and Fondecyt-Regular #1210606.

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
