# A novel notion of barycenter for probability distributions based on optimal weak mass transport

ELSA CAZELLES, FELIPE TOBAR, JOAQUÍN FONTBONA

## A   Additional mathematical background

### A.1   $p$-Wasserstein distance

For $p = 2$ and $\mu, \nu \in \mathcal{P}_2(\mathbb{R}^d)$ such that $\mu$ is absolutely continuous (*a.c.*) with respect to Lebesgue measure, the unique optimal plan is concentrated on the graph of a measurable map and Eq. (1) boils down to Monge's problem:

$$W_2(\mu, \nu) = \left( \min_{T \in \mathbb{T}(\mu,\nu)} \int_{\mathbb{R}^d \times \mathbb{R}^d} \|x - T(x)\|^2 \mathrm{d}\mu(x) \right)^{1/2}, \tag{14}$$

where $\mathbb{T}(\mu, \nu)$ is the set of measurable functions $T : \mathbb{R}^d \to \mathbb{R}^d$ such that $\nu = T\#\mu$. The *pushforward* operator $\#$ is defined such that for any measurable set $B \subset \mathbb{R}^d$, we have $\nu(B) = \mu(T^{-1}(B))$. In such a case, the optimal measurable map $T$ in Eq. (14) is uniquely defined (see *e.g.* Th. 9.4 in [42]) and called *Monge map*.

### A.2   Continuity of $V$

**Theorem 6** ([9], Theorem 1.5). *Let $(\mu_n)_n \subset \mathcal{P}_2(\mathbb{R}^d)$ and $(\nu_n)_n \subset \mathcal{P}_1(\mathbb{R}^d)$. Then*

$$\begin{cases} \mu_n \to \mu & in\ W_2 \\ \nu_n \to \nu & in\ W_1 \end{cases} \implies \lim_n V(\mu_n | \nu_n) = V(\mu | \nu).$$

### A.3   On the barycentric projection

For a given transport plan $\pi \in \Pi(\mu, \nu)$, with $\mu, \nu \in \mathcal{P}_2(\mathbb{R}^d)$, the associated barycentric projection is given by

$$S : x \mapsto \int_{\mathbb{R}^d} y \mathrm{d}\pi_x(y).$$

First, for each $x \in \mathbb{R}^d$, $S(x)$ realises $\min_z \mathbb{E}_{Y \sim \pi_x}(\|z - Y\|^2)$. Second, this barycentric map $S$ is actually optimal for the Monge's problem Eq. (14) between $\mu$ and $S\#\mu$, by Theorem 2.

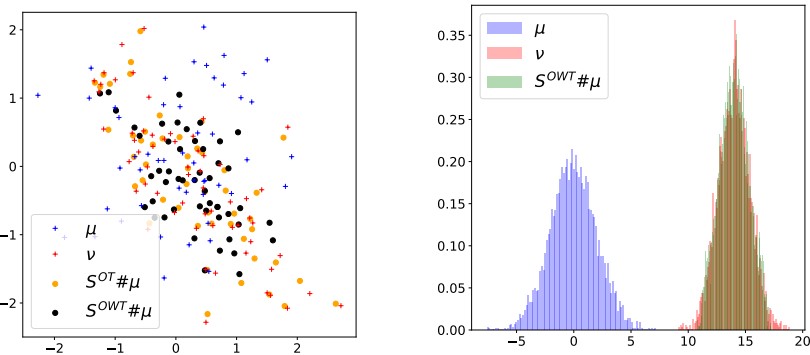

Figure 6: Example of pushforward measures constructed from barycentric projections for two measures $\mu$ and $\nu$ in two dimensions (left) and one dimension (right).

We next illustrate the differences between the optimal barycentric map and a barycentric map constructed from an OT plan in the classical Kantorovich formulation in Eq. (1). We sampled $r = 50$ observations $X_i$ and $m = 60$ observations $Y_i$, each sets from a 2D Gaussian. We then defined the source and target distributions as $\mu = \frac{1}{r}\sum \delta_{X_i}$ and $\nu = \frac{1}{m}\sum \delta_{Y_i}$ respectively. Figure 6(left), shows these discrete distributions together with the pushforward measures $S^{OWT}\#\mu$ and $S^{OT}\#\mu$ constructed from the optimal weak plan $\pi^{OWT}$ and an optimal plan $\pi^{OT}$ respectively. The measure $S^{OT}\#\mu$ reasonably fits the target distribution $\nu$, since when $\mu$ is $a.c.$, $S^{OT}\#\mu = \nu$. In particular, if $\mu$ and $\nu$ had the same number of points, $S^{OT}\#\mu$ would have matched $\nu$. Regarding the measure $S^{OWT}\#\mu$, recall that $V(\mu|\nu) = \inf_{\eta \leq_c \nu} W_2^2(\mu, \eta) = W_2^2(\mu, S^{OWT}\#\mu)$, and therefore $S^{OWT}\#\mu \leq_c \nu$. Lastly, we have $W_2^2(\mu, \nu) = 0.85$, and $V(\mu|\nu) = 0.52 \leq W_2^2(\mu, S^{OT}\#\mu) = 0.81$ as expected.

In Figure 6(right), we present an example in one dimension, where we sample 4000 observations from $\mathcal{N}(0, 2)$ (resp. $\mathcal{N}(14, 1.4)$) to construct the empirical source measure $\mu$ (resp. empirical target measure $\nu$). The distributions $\mu$ and $\nu$ are presented in the form of histograms. The distribution resulting from the optimal weak transport map $S_\mu^\nu\#\mu$ is in convex order with $\nu$.

## B  Proofs of Section 3

*Proof of Proposition 1.* Let $(\mu_m)_m \subset \mathcal{P}_2(\mathbb{R}^d)$ be a minimising sequence of $F(\mu) := \sum_{i=1}^n \lambda_i V(\mu|\nu_i)$ and let $M < \infty$ be such that $F(\mu_m) \leq M$ for all $m$. Then $(\mu_m)_m$ is tight. Indeed,

$$\int \|x\|^2 d\mu_m(x) \leq 2\sum_{i=1}^n \lambda_i \inf_{\pi \in \Pi(\mu_m, \nu_i)} \left[ \int \|x - \int y d\pi_x(y)\|^2 d\mu_m(x) + \int \|\int y d\pi_x(y)\|^2 d\mu_m(x) \right]$$

$$\leq 2M + 2\iint \|y\|^2 d\pi_x(y) d\mu_m(x) \leq 2M + 2\sum_{i=1}^n \lambda_i \int \|y\|^2 d\nu_i(y),$$

where the second inequality comes from Jensen's inequality. By Prokhorov's theorem, there exists a subsequence still denoted $(\mu_m)_m$ that weakly converges toward a probability measure $\mu^*$. Recall that $\mu$ belongs to $\mathcal{P}_2(\mathbb{R}^d)$ since $\|\cdot\|^2$ is a l.s.c. function bounded from below and therefore $\int \|x\|^2 d\mu(x) \leq \liminf_m \int \|x\|^2 d\mu_m(x) < \infty$. By Theorem 6, we have that

$$F(\mu^*) = \sum_{i=1}^n \lambda_i \lim_m V(\mu_m|\nu_i) = \lim_m F(\mu_m) = \min_{\mu \in \mathcal{P}_2(\mathbb{R}^d)} F(\mu),$$

thus $F$ admits at least a minimiser. $\qquad\square$

*Proof of Lemma 1.* By Strassen's theorem, we can build $X' \sim \mu'$ and $X \sim \mu$ in the same probability space, in such a way that $\mathbb{E}(X|X') = X'$. Denote by $\eta^*$ the law $\eta$ attaining $\inf_{\eta \leq_c \nu} W_2^2(\mu, \eta)$, and let $(X, Z) = (X, S_\mu^\nu(X))$ be the realisation of the optimal coupling for $W_2$ of $\mu$ and $\eta^*$, which can also be constructed in the same probability space due to its specific form. Then, by (the conditional version of) Jensen's inequality we have

$$V(\mu|\nu) = W_2^2(\mu, \eta^*) = \mathbb{E}\left[\mathbb{E}\left(\|X - S_\mu^\nu(X)\|^2 | X'\right)\right] \geq \mathbb{E}\|X' - \mathbb{E}(S_\mu^\nu(X)|X')\|^2.$$

Recall now that $S_\mu^\nu(X) = \mathbb{E}(Y|X)$, where the conditional expectation is a measurable function only of $X$, constructed from the joint law $\pi^{\mu,\nu}$. Thus, for every nonnegative convex function $\phi$, by applying twice Jensen's inequality we get

$$\mathbb{E}\phi(\mathbb{E}(S_\mu^\nu(X)|X')) \leq \mathbb{E}\phi(S_\mu^\nu(X)) = \mathbb{E}\phi(\mathbb{E}(Y|X)) \leq \mathbb{E}\phi(Y),$$

where $Y \sim \nu$. That is to say, the law $\eta$ of the r.v. $\mathbb{E}(S_\mu^\nu(X)|X')$ satisfies $\eta \leq_c \nu$. It follows that

$$V(\mu|\nu) \geq W_2^2(\mu', \eta) \geq \inf_{\tilde\eta \leq_c \nu} W_2^2(\mu', \tilde\eta) = V(\mu'|\nu).$$

This immediately implies that $\mu'$ is a weak barycenter whenever $\mu$ is. In particular, if $\mu$ is a weak barycenter, then so is the Dirac mass supported on its mean. We then deduce that the set of minimisers of $\sum_i \lambda_i V(\mu|\nu_i)$ admits at least a Dirac mass $\delta_\omega$ and

$$V(\delta_\omega|\nu_i) = \int \|x - \int y d\pi_x(y)\|^2 d\delta_\omega(x) = \|\omega - \mathbb{E}Y_i\|^2.$$

This implies that $\inf_\omega \sum \lambda_i V(\delta_\omega|\nu_i)$ is uniquely attained for $\bar\omega = \sum \lambda_i \mathbb{E}Y_i$. $\qquad\square$

*Proof of Proposition 2.* A probability measure $\mu$ is a weak barycenter if and only if $\sum_{i=1}^{n} \lambda_i V(\mu|\nu_i)$ is equal to the r.h.s. of (7). Let us suppose first that $\mathbb{E}Y_i = m$ for all $1 \leq i \leq n$, in which case the infimum in (5) is equal to 0. Then, $\mu$ is a weak barycenter if and only if $\mu \leq_c \nu_i$ for all $1 \leq i \leq n$ by definition of weak optimal transport (2), since in this case $V(\mu|\nu_i) = 0$. The general case can be reduced to the previous one, noting that

$$
\begin{aligned}
V(\mu|\nu_i) &= \inf_{\eta \leq_c \nu_i} W_2^2(\mu, \eta) \\
&= \inf_{\eta \leq_c \hat{\nu}_i} W_2^2(\hat{\mu}, \eta) + \|\mathbb{E}_\mu(X) - \mathbb{E}_{\nu_i}(Y_i)\|^2 \\
&= V(\hat{\mu}|\hat{\nu}_i) + \|\mathbb{E}_\mu(X) - \mathbb{E}_{\nu_i}(Y_i)\|^2,
\end{aligned}
$$

so that minimising $\sum_{i=1}^{n} \lambda_i V(\mu|\nu_i)$ over $\mu \in \mathcal{P}(\mathbb{R}^d)$ is equivalent to minimising $\sum_{i=1}^{n} \lambda_i V(\mu'|\hat{\nu}_i) + \sum_{i=1}^{n} \lambda_i \|\omega - \mathbb{E}_{\nu_i} Y_i\|^2$ over the two independent parameters $(\omega, \mu')$, with $\omega \in \mathbb{R}^d$ and $\mu' \in \mathcal{P}(\mathbb{R}^d)$ centered, taking $\mu$ as the law of $X = X' + \omega$ with $X' \sim \mu'$. $\qquad \square$

*Proof of Lemma 2.* Thanks to Prop. 3.3 in [5], we have that the (unique) Wasserstein barycenter verifies $\tilde{\mu} = \left( \sum_{i=1}^{n} \lambda_i T_{\tilde{\mu}}^{\nu_i} \right) \#\tilde{\mu}$ where $T_{\tilde{\mu}}^{\nu_i}$ is the optimal Monge map between $\tilde{\mu}$ and $\nu_i$ (see (14)). Moreover, from Proposition 4, a weak barycenter $\bar{\mu}$ also checks $\bar{\mu} = \left( \sum_{i=1}^{n} \lambda_i S_{\bar{\mu}}^{\nu_i} \right) \#\bar{\mu}$, where $S_{\bar{\mu}}^{\nu_i}$ is the optimal barycentric projection associated to $\bar{\pi}^i$ for $V(\bar{\mu}|\nu_i)$. Therefore, by Jensen's inequality applied twice,

$$
\begin{aligned}
W_2^2(\bar{\mu}, \tilde{\mu}) &\leq \iint \|x - y\|^2 \mathrm{d}\bar{\mu}(x)\mathrm{d}\tilde{\mu}(y) = \iint \|\sum_{i=1}^{n} \lambda_i S_{\bar{\mu}}^i(x) - \sum_{i=1}^{n} \lambda_i T_{\tilde{\mu}}^i(y)\|^2 \mathrm{d}\bar{\mu}(x)\mathrm{d}\tilde{\mu}(y) \\
&\leq \sum_{i=1}^{n} \lambda_i \iiint \|T_{\tilde{\mu}}^i(y) - z\|^2 \mathrm{d}\bar{\pi}_x^i(z)\mathrm{d}\bar{\mu}(x)\mathrm{d}\tilde{\mu}(y) = \sum_{i=1}^{n} \lambda_i \iint \|T_{\tilde{\mu}}^i(y) - z\|^2 \mathrm{d}\tilde{\mu}(y)\mathrm{d}\bar{\pi}^i(x, z) \\
&= \sum_{i=1}^{n} \lambda_i \iint \|y - z\|^2 \mathrm{d}\nu_i(y)\mathrm{d}\nu_i(z) = 2\sum_{i=1}^{n} \lambda_i \iint \left( \mathbb{E}\|Y_i\|^2 - \|\mathbb{E}Y_i\|^2 \right).
\end{aligned}
$$

$\qquad \square$

*Proof of Theorem 3.* Observe first that, by Theorem 2 and Strassen's theorem, solving the OWT problem (2) provides a unique (in law) coupling of three random variables $(X, Y, Z)$ such that:

i) $(X, Y)$ has joint law $\pi^{\mu,\nu}$; in particular $X$ and $Y$ have the laws $\mu$ and $\nu$ respectively,

ii) $Z = S_\mu^\nu(X) = \mathbb{E}(Y|X)$ a.s., it has law $\eta^*$ and it is optimally coupled to $X$ in the sense of the optimal transport problem (1),

iii) $(Z, Y)$ is a martingale, that is $\mathbb{E}(Y|Z) = Z$ a.s..

Bringing all together we get the decomposition:

$$
Y = Z + Y - Z = S_\mu^\nu(X) + Y - \mathbb{E}(Y|X). \tag{15}
$$

Now, by Lemma 1, if $X \sim \mu$ then the Dirac mass $\delta_{\mathbb{E}X}$ is a weak barycenter too. Thus we have on one hand:

$$
\sum_{i=1}^{n} \lambda_i V(\mu|\nu_i) = \sum_{i=1}^{n} \lambda_i V(\delta_{\mathbb{E}X}|\nu_i). \tag{16}
$$

Using Jensen's inequality, we see on the other hand that

$$
\begin{aligned}
V(\mu|\nu_i) &= \inf_{\eta \leq_c \nu_i} W_2^2(\mu, \eta) \\
&= \inf_{\eta \leq_c \nu_i} \mathbb{E}\|X - Z\|^2, \text{ with } (X, Z) \text{ an optimal coupling for } W_2^2 \text{ of } \mu \text{ and } \eta \\
&\geq \inf_{\eta \leq_c \nu_i} \|\mathbb{E}X - \mathbb{E}Z\|^2 = \inf_{\eta \leq_c \nu_i} W_2^2(\delta_{\mathbb{E}X}, \delta_{\mathbb{E}Z}) \\
&\geq \inf_{\tilde{\eta} \leq_c \nu_i} W_2^2(\delta_{\mathbb{E}X}, \tilde{\eta}) = V(\delta_{\mathbb{E}X}|\nu_i).
\end{aligned}
$$

Identity (16) thus implies $V(\mu|\nu_i) = V(\delta_{\mathbb{E}X}|\nu_i)$ for all $i$. Denoting by $\eta_i$ the law $\eta$ attaining $\inf_{\eta \leq_c \nu_i} W_2^2(\mu, \eta)$, and by $(X, Z_i)$ the optimal coupling for $W_2$ of $\mu$ and $\eta_i$, we see that the latter can only occur if the equality case $\mathbb{E}\|X - Z_i\|^2 = \|\mathbb{E}X - \mathbb{E}Z_i\|^2$ in Jensen's inequality holds. This implies that $X - Z_i$ is deterministic for each $i$. Since $\mathbb{E}Z_i = \mathbb{E}Y_i$, we thus must have $X - Z_i = \mathbb{E}X - \mathbb{E}Y_i$. Taking $Z = Z_i$ and $Y = Y_i$ in Eq. (15), and noting that $S_\mu^{\nu_i}(X) = Z_i = X - (\mathbb{E}X - \mathbb{E}Y_i)$ the statement follows. $\qquad\square$

## C  Proofs of Section 3.3

*Proof of Lemma 3.* Let $(\mu_m)$ and $(\nu_m)$, $m \in \mathbb{N}$, be two sequences in $\mathcal{P}_2(\mathbb{R}^d)$ respectively converging to $\mu$ and $\nu$ w.r.t. $W_2$. Then, $(\mu_m)$ and $(\nu_m)$ are tight and thus the sequence $(\pi^m) := (\pi^{\mu_m, \nu_m})$ is tight too. Let $\pi^{m_k}$ be a weakly convergent subsequence and $\pi$ its limit. By Proposition 2.8 in [8] we have

$$\liminf_m V(\mu_m|\nu_m) = \liminf_m \int \|x - \int y \mathrm{d}\pi_x^{m_k}\|^2 \mathrm{d}\mu_{m_k}(x) \geq \int \|x - \int y \mathrm{d}\pi_x\|^2 \mathrm{d}\mu(x) \geq V(\mu|\nu).$$

However, we have $\lim_m V(\mu_m|\nu_m) = V(\mu|\nu)$ thanks to Theorem 6, hence $\int \|x - \int y \mathrm{d}\pi_x\|^2 \mathrm{d}\mu(x) = V(\mu|\nu)$. By uniqueness of the optimum for problem (2) we deduce that $\pi = \pi^{\mu,\nu}$. Since the same holds true for any weak limiting point of $(\pi^m)$, it follows that $\pi^m$ weakly converges to $\pi^{\mu,\nu}$. Last, since $\int \|x\|^2 + \|y\|^2 \, \mathrm{d}\pi^m(x,y) = \int \|x\|^2 \mathrm{d}\mu_m(x) + \int \|y\|^2 \mathrm{d}\nu_m(y)$, this quantity converges to $\int \|x\|^2 \mathrm{d}\mu(x) + \int \|y\|^2 \mathrm{d}\nu(y) = \int \|x\|^2 + \|y\|^2 \, \mathrm{d}\pi(x,y)$, whence $W_2(\pi^n, \pi) \to 0$, and $(\mu, \nu) \in (\mathcal{P}_2(\mathbb{R}^d))^2 \mapsto \pi^{\mu,\nu} \in \mathcal{P}_2(\mathbb{R}^d \times \mathbb{R}^d)$ is continuous, as required (hence measurable).

We now establish the joint measurability of $(x, \nu) \in \mathbb{R}^d \times \mathcal{P}_2(\mathbb{R}^d) \mapsto S_\mu^\nu(x)$ for fixed $\mu$. Notice this is a stronger statement than just measurability in the $x$ variable, for each $(\mu, \nu)$. Write $\bar{B}(x, r)$ for the closed ball of radius $r > 0$ centered at $x$. One easily checks that the function

$$(x, \pi) \mapsto \Psi_r(x, \pi) := \frac{\int y \mathbf{1}_{\{(y,z):z \in \bar{B}(x,r)\}} \mathrm{d}\pi(z, y)}{\int \mathbf{1}_{\{(y,z):z \in \bar{B}(x,r)\}} \mathrm{d}\pi(z, y)}$$

is measurable w.r.t. the pair $(x, \pi)$, the two integrals being limits of integrals with respect to $\mathrm{d}\pi(z, y)$, of some bounded continuous functions of $(x, y, z)$. Thus, $\limsup_{r \to 0} \Psi_r(x, \pi)$, $\liminf_{r \to 0} \Psi_r(x, \pi)$ and the function $\Phi(x, \pi) := \limsup_{r \to 0} \Psi_r(x, \pi) \mathbf{1}_{\{\limsup_{r \to 0} \Psi_r(x,\pi) = \liminf_{r \to 0} \Psi_r(x,\pi)\}}$ depend in a measurable way on $(x, \pi)$. It follows that $(x, \mu, \nu) \mapsto \Phi(x, \pi^{\mu,\nu})$ is measurable as the composition of two measurable functions. But notice that for each fixed $\mu \in \mathcal{P}_2(\mathbb{R}^d)$ one has $\Psi_r(x, \pi^{\mu,\nu}) = \frac{\int_{\bar{B}(x,r)} [\int y \mathrm{d}\pi_z(y)] \mathrm{d}\mu(z)}{\mu(\bar{B}(x,r))}$ which, by the Lebesgue derivation theorem for Radon measures (*see e.g.* [16]), converges $\mathrm{d}\mu(x)$ a.s. in $x$, to $\int y \mathrm{d}\pi_x(y) = S_\mu^\nu(x)$. Thus, for each $\mu \in \mathcal{P}_2(\mathbb{R}^d)$,

$$S_\mu^\nu(x) = \Phi(x, \pi^{\mu,\nu}) \quad \text{for all } \nu \in \mathcal{P}_2(\mathbb{R}^d) \text{ and } \mathrm{d}\mu(x) \text{ a.e. } x,$$

with $(x, \nu) \mapsto \Phi(x, \pi^{\mu,\nu})$ a measurable function. The conclusion follows.

$\qquad\square$

*Proof of Proposition 3.* By Theorem 6.16 in [42], we know that their exists a sequence of discretely supported distributions $(\mathbb{Q}_n)_n \subset \mathcal{P}_2(\mathcal{P}_2(\mathbb{R}^d))$ of the form $\mathbb{Q}_n = \sum_{i=1}^n \lambda_i \delta_{\nu_i}$, with $(\lambda_i)_{1 \leq i \leq n}$ in the simplex, and such that $W_2^2(\mathbb{Q}, \mathbb{Q}_n) := \inf_{\pi \in \Pi(\mathbb{Q}, \mathbb{Q}_n)} \int W_2^2(\nu, \tilde{\nu}) \mathrm{d}\pi(\nu, \tilde{\nu}) \to 0$. We set

$$L_n(\mu) := \int_{\mathcal{P}_2(\mathbb{R}^d)} V(\mu|\nu) \mathrm{d}\mathbb{Q}_n(\nu) = \sum_{i=1}^n \lambda_i V(\mu|\nu_i).$$

We denote $\mu^n \in \mathcal{P}_2(\mathbb{R}^d)$ the minimiser of $L_n$. Let us prove that $(\mu^n)_n$ is tight. First, $\mu^n$ admits moments of order 2 thanks to Jensen's inequality:

$$\int \|x\|^2 \mathrm{d}\mu^n(x) \leq \sum_{i=1}^n \lambda_i \left[ \int \|x - S_{\mu^n}^{\nu_i}(x)\|^2 \mathrm{d}\mu^n(x) + \int \|S_{\mu^n}^{\nu_i}(x)\|^2 \mathrm{d}\mu^n(x) \right]$$

$$\leq \sum_{i=1}^n \lambda_i V(\mu^n|\nu_i) + \sum_{i=1}^n \lambda_i \int \|y\|^2 \mathrm{d}\nu_i(y)$$

$$\leq \sum_{i=1}^n \lambda_i V(\mu|\nu_i) + \sum_{i=1}^n \lambda_i \int \|y\|^2 \mathrm{d}\nu_i(y) \quad \text{for some } \mu \in \mathcal{P}_2(\mathbb{R}^d) \text{ since } \mu^n \text{ minimises } L_n$$

$$\leq 2 \int \|x\|^2 \mathrm{d}\mu(x) + 3 \sum_{i=1}^n \lambda_i \int \|y\|^2 \mathrm{d}\nu_i(y),$$

where the last inequality comes from $V(\mu|\nu_i) = \int \|x - S_\mu^{\nu_i}(x)\|^2 \mathrm{d}\mu(x) \leq 2\int \|x\|^2 \mathrm{d}\mu(x) + 2\int \|S_\mu^{\nu_i}(x)\|^2 \mathrm{d}\mu(x)$. Moreover, since $W_2^2(\mathbb{Q}, \mathbb{Q}_n) \to 0$, we have (Lemma 5.1.7 in [6]) that $\int \psi(\nu)\mathrm{d}\mathbb{Q}_n(\nu) \to \int \psi(\nu)\mathrm{d}\mathbb{Q}(\nu)$ for any function $\psi$ such that $|\psi(\nu)| \leq a + bW_2^2(\nu, \nu_0), a, b \geq 0$. In particular, choosing $\psi(\nu) = W_2^2(\nu, \delta_0) = \int \|y\|^2 \mathrm{d}\nu(y)$, it implies that $\sum_{i=1}^n \lambda_i \int \|y\|^2 \mathrm{d}\nu_i(y) \to \int \int \|y\|^2 \mathrm{d}\nu(y)\mathrm{d}\mathbb{Q}(\nu) < \infty$. Therefore $\left(\sum_{i=1}^n \lambda_i \int \|y\|^2 \mathrm{d}\nu_i(y)\right)_n$ is bounded and $(\mu^n)_n$ is tight. Thus by Prokhorov's theorem, there exists a subsequence, still denoted $(\mu^n)_n$, that converges towards $\bar{\mu}$.

Let us now prove that this particular $\bar{\mu}$ minimises the function $L : \mu \mapsto \int_{\mathcal{P}_2(\mathbb{R}^d)} V(\mu|\nu)\mathrm{d}\mathbb{Q}(\nu)$. First, let $\eta \in \mathcal{P}_2(\mathbb{R}^d)$, still by Lemma 5.1.7 in [6] and since $V(\eta|\nu) \leq W_2^2(\eta, \nu)$, we get that $L(\eta) = \int V(\eta|\nu)\mathrm{d}\mathbb{Q}(\nu) \geq \liminf_{n\to\infty} \int V(\eta|\nu)\mathrm{d}\mathbb{Q}_n(\nu)$. Since for each $n$, the distribution $\mu^n$ minimises $L_n$, we have

$$\liminf_{n\to\infty} \int V(\eta|\nu)\mathrm{d}\mathbb{Q}_n(\nu) \geq \liminf_{n\to\infty} \int V(\mu^n|\nu)\mathrm{d}\mathbb{Q}_n(\nu). \tag{17}$$

Thanks to Fatou's Lemma for sequences of measures $(\mathbb{Q})_n$ (see [22]), we have that

$$\liminf_{n\to\infty} \int V(\mu^n|\nu)\mathrm{d}\mathbb{Q}_n(\nu) \geq \int \liminf_{n\to\infty} V(\mu^n|\nu)\mathrm{d}\mathbb{Q}(\nu) = \int V(\bar{\mu}|\nu)\mathrm{d}\mathbb{Q}(\nu),$$

where the last equality comes from the lower semi-continuity of $V$ (Theorem 2.9 in [8]). This proves that $\bar{\mu}$ minimises $L$. $\square$

## D    Proofs of Section 4

The proof of Theorem 4, on the continuity of $G : \mu \mapsto \left(\sum_{i=1}^n \lambda_i S_\mu^{\nu_i}\right) \#\mu$, leans on the two following technical lemmas.

**Lemma 5.** *Let $(\rho_m)_m$ be a given sequence and $\nu$ be a fixed law in $\mathcal{P}_2(\mathbb{R}^d)$. For each $m$, let $S_m := S_{\rho_m}^\nu$ denote the barycenter map associated with the optimal coupling $\pi^{\rho_m, \nu}$ for (2). Then, the sequence of laws $(S_m \#\rho_m)_m$ has uniformly integrable second moments.*

*Proof of Lemma 5.* Let $(X_m, Y_m)$ be a pair of random variables (r.v.) with joint law $\pi^{\mu_m, \nu}$, defined on some probability space $(\Omega, \mathcal{F}, \mathbb{P})$. Notice that $S_m \#\rho_m$ is the law of the r.v. $\mathbb{E}(Y_m|X_m)$. Then, for each $M, K \geq 0$ we have

$$\int_{\{\|x\|^2 \geq M\}} \|x\|^2 \mathrm{d}S_m \#\rho_m(x) = \mathbb{E}(\|\mathbb{E}(Y_m|X_m)\|^2 \mathbf{1}_{\{\|\mathbb{E}(Y_m|X_m)\|^2 \geq M\}})$$

$$\leq \mathbb{E}(\mathbb{E}(\|Y_m\|^2|X_m)\mathbf{1}_{\{\|\mathbb{E}(Y_m|X_m)\|^2 \geq M\}})$$

$$= \mathbb{E}(\|Y_m\|^2 \mathbf{1}_{\{\|\mathbb{E}(Y_m|X_m)\|^2 \geq M, \|Y_m\|^2 \geq K\}})$$

$$+ \mathbb{E}(\|Y_m\|^2 \mathbf{1}_{\{\|\mathbb{E}(Y_m|X_m)\|^2 \geq M, \|Y_m\|^2 < K\}})$$

$$\leq \mathbb{E}(\|Y_m\|^2 \mathbf{1}_{\|Y_m\|^2 \geq K}) + \frac{K}{M}\mathbb{E}(\|\mathbb{E}(Y_m|X_m)\|^2),$$

where we have used Jensen's inequality and the fact that $\mathbb{E}(Y_m|X_m)$ is measurable w.r.t. the $\sigma$-field generated by $X_m$. Applying Jensen's inequality to the last term again, and recalling that $Y_m$ has law $\nu \in \mathcal{P}_2(\mathbb{R}^d)$, we deduce that

$$\sup_m \int_{\{\|x\|^2 \geq M\}} \|x\|^2 \mathrm{d}S_m \# \rho_m(x) \leq \int_{\{\|x\|^2 \geq K\}} \|x\|^2 \mathrm{d}\nu(x) + \frac{K}{M} \int \|x\|^2 \mathrm{d}\nu(x), \quad (18)$$

which is smaller than a given $\varepsilon > 0$, by choosing $K > 0$ and then $M > 0$ large enough. $\qquad\square$

**Lemma 6.** *Let $(\rho_m)_n, \rho$ in $\mathcal{P}_2(\mathbb{R}^d)$ be such that $W_2(\rho_m, \rho) \to 0$. We have:*

   i) *For each $\nu \in \mathcal{P}_2(\mathbb{R}^d)$ the sequence of laws $((id, S_{\rho_m}^\nu)\#\rho_m)_m$ converges w.r.t. $W_2$ in $\mathcal{P}_2(\mathbb{R}^d \times \mathbb{R}^d)$ to $(id, S_\rho^\nu)\#\rho$.*

   ii) *There exists in some probability space $(\Omega, \mathcal{F}, \mathbb{P})$, a sequence of r.v. $(X_m)_m$ of laws $(\rho_m)_m$ and a r.v. $X$ of law $\rho$ such that, for each $\nu \in \mathcal{P}_2(\mathbb{R}^d)$, the sequence $(X_m, S_{\rho_m}^\nu(X_m))_m$ (with laws $((id, S_{\rho_m}^\nu)\#\rho_m)_m$) converges in $\mathbb{L}^2(\Omega, \mathcal{F}, \mathbb{P})$ to $(X, S_\rho^\nu(X))$ (with law $(id, S_\rho^\nu)\#\rho$).*

*Proof of Lemma 6.* For the entire proof, we fix a $\nu \in \mathcal{P}_2(\mathbb{R}^d)$ and we write $S_m := S_{\rho_m}^\nu$ and $S := S_\rho^\nu$ for simplicity.

i) By Theorem 2 and Part 1. of Theorem 1.5 in [9], $(S_m \# \rho_m)_m$ converges to $S \# \rho$ w.r.t. $W_1$ and, by Lemma 5, also with respect to $W_2$. In particular, the sequence $((id, S_m)\#\rho_m)_m$ has tight marginals, and therefore it is tight too.

Let us identify its weak limiting points. For simplicity we rename $((id, S_m)\#\rho_m)_m$ a weakly convergent subsequence. By the previous discussion, its weak limit $\mathrm{d}\hat{\rho}(x, z)$ clearly has first and second marginal laws equal to $\mathrm{d}\rho(x)$ and $\mathrm{d}S\#\rho(z)$ respectively. Moreover, $\int \|x\|^2 \mathrm{d}\rho_m(x) + \int \|z\|^2 \mathrm{d}S_m\#\rho_m(z) \to \int \|x\|^2 + \|z\|^2 \mathrm{d}\hat{\rho}(x, z)$, hence $((id, S_m)\#\rho_m)_m$ converges to some $\hat{\pi}$ with respect to $W_2$ in $\mathcal{P}_2(\mathbb{R}^d \times \mathbb{R}^d)$.

Now, by the characterisation of optimisers in Theorem 2, we have $V(\rho_m|\nu) = W_2^2(\rho_m, S_m\#\rho_m) = \int \|x - S_m(x)\|^2 \mathrm{d}\rho_m(x)$. Taking $m \to \infty$, and thanks to Theorem 6, we finally obtain

$$V(\rho|\nu) = W_2^2(\rho, S\#\rho) = \int \|x - z\|^2 \mathrm{d}\hat{\pi}(x, z).$$

In particular, using again Theorem 2, we conclude that $\mathrm{d}\hat{\pi}(x, z)$ must be of the form $(id, S)\#\rho$.

ii) By Skorohod's representation theorem, one can construct simultaneously in some probability space $(\Omega, \mathcal{F}, \mathbb{P})$, a sequence of r.v. $(X_m)_m$ of laws $(\rho_m)_m$ and a r.v. $X$ of law $\rho$ such that $(X_m)_m$ converges $\mathbb{P}-$ a.s. to $X$. Moreover, since the sequence $(\rho_m)_m$ converges w.r.t. $W_2$ in $\mathcal{P}_2(\mathbb{R}^d)$, it has uniformly integrable second order moments. It follows that the sequence of r.v. $(|X_m|^2)_m$ is uniformly integrable and, by the Vitali convergence theorem, that $(X_m)_m$ also converges to $X$ in $\mathbb{L}^2(\Omega, \mathcal{F}, \mathbb{P})$.

Now, by Lemma 5, the sequence of r.v. $(|S_m(X_m)|^2)_n$ is uniformly integrable too. Thus, by the Vitali convergence theorem, the statement will follow by proving that $S_m(X_m)$ converges in $\mathbb{P}-$probability to $S(X)$.

For each $N \in \mathbb{N}$, let $y \mapsto (y)^N$ denote the truncation of a vector $y \in \mathbb{R}^d$ obtained by projecting it onto the centered ball of radius $N$, $(y)^N := (1 \wedge \frac{N}{|y|})y$, which is a $1-$Lipschitz function bounded by $N$. By Theorem 2, the functions $S_m^N := (S_m)^N$ are then $1-$Lipschitz and bounded uniformly in $m \in \mathbb{N}$. Therefore, by the Arzela-Ascoli theorem, their restrictions to each compact cylinder set $R$ of $\mathbb{R}^d$ defines a relatively compact set of functions, with respect to the uniform topology in $C(R, \mathbb{R}^d)$. It follows by a diagonal argument that some subsequence $(S_{m_k}^N)_k$ converges, uniformly on compact sets, to some continuous function $\tilde{S}$ on $\mathbb{R}^d$. Since $X_n$ converges a.s. to the finite value $X$, we deduce that $\mathbb{P}-$a.s. as $k \to \infty$,

$$(X_{m_k}, S_{m_k}^N(X_{m_k})) \to (X, \tilde{S}(X)).$$

Notice now that $(X_{m_k}, S_{m_k}^N(X_{m_k}))$ has the law $(id, (\cdot)^N \circ S_{m_k})\#\rho_{m_k}$ for each $k$ and thus, by part a) and continuity of the mapping $(x, y) \mapsto (x, (y)^N)$, the r.v. $(X, \tilde{S}(X))$ has the law $(id, (\cdot)^N \circ S)\#\rho$. Hence we deduce that

$$(X, \tilde{S}(X)) = (X, (S(X))^N)$$

$\mathbb{P}-$ almost surely. The previous arguments can be applied not just to $(X_m)_m$ but to any subsequence of it. That is, we can similarly prove that any subsequence of $(X_m, (S_m(X_m))^N)_m$ has a subsequence that a.s. converges to $(X, (S(X))^N)$. This means that, for each $N \in \mathbb{N}$

$$(X_m, (S_m(X_m))^N) \to (X, (S(X))^N)$$

in $\mathbb{P}-$ probability when $n \to \infty$. To conclude, by tightness we can find for each $\eta > 0$ some $N \in \mathbb{N}$ large enough so that $\mathbb{P}(|S(X)| \geq N) \leq \eta$ and $\mathbb{P}(|S_m(X_m)| \geq N) \leq \eta$ for all $m \in \mathbb{N}$, which yields for each $\varepsilon > 0$,

$$\mathbb{P}(|S_m(X_m) - S(X)| \geq \varepsilon) \leq 2\eta + \mathbb{P}(|(S_m(X_m))^N - (S(X))^N| \geq \varepsilon).$$

Thus $\limsup_m \mathbb{P}(|S_m(X_m) - S(X)| \geq \varepsilon) \leq 2\eta$ for arbitrary $\eta > 0$ or, equivalently, $\mathbb{P}(|S_m(X_m) - S(X)| \geq \varepsilon) \to 0$ as $m \to \infty$, which concludes the proof of b). $\square$

We can now proceed to the proof of continuity of $G : \mu \mapsto \left(\sum_{i=1}^n \lambda_i S_\mu^{\nu_i}\right) \#\mu$.

*Proof of Theorem 4.* Let $(\rho_m)_m, \rho$ in $\mathcal{P}_2(\mathbb{R}^d)$ such that $W_2(\rho_m, \rho) \to 0$. We need to prove that $W_2^2(G(\rho_m), G(\rho)) \to 0$. For each $m$, we write $S_m^i := S_{\rho_m}^{\nu_i}$ and $S^i := S_\rho^{\nu_i}$.

By Lemma 6.ii), there exists in some probability space a sequence $(X_m)_m$ of laws $(\rho_m)_m$ and a r.v. $X$ of law $\rho$ such that

$$(S_m^1(X_m), ..., S_m^n(X_m)) \to (S^1(X), ..., S^n(X)) \quad \text{in} \quad \mathbb{L}^2(\mathbb{P}).$$

Therefore, $\sum_{i=1}^n \lambda_i S_m^i(X_m)$ converges to $\sum_{i=1}^n \lambda_i S^i(X)$ in $\mathbb{L}^2(\mathbb{P})$. Since $\sum_{i=1}^n \lambda_i S_m^i(X_m)$ has law $G(\rho_m)$ and $\sum_{i=1}^n \lambda_i S^i(X)$ has law $G(\rho)$, the proof is complete. $\square$

*Proof of Proposition 4.* As in [5], we easily see that

$$\sum_{i=1}^n \lambda_i \int \|x - S_\mu^{\nu_i}(x)\|^2 d\mu(x) = \sum_{i=1}^n \lambda_i \int \|\bar{S}(x) - S_\mu^{\nu_i}(x)\|^2 d\mu(x) + \int \|x - \bar{S}(x)\|^2 d\mu(x).$$

But $\int \|x - S_\mu^{\nu_i}(x)\|^2 d\mu(x) = W_2^2(\mu, S_\mu^{\nu_i}\#\mu)$ since from Thm 1.4 in [8] the barycentric map $S_\mu^{\nu_i}$ is an optimal map for the Monge problem between $\mu$ and $S_\mu^{\nu_i}\#\mu$. Moreover, by definition $G(\mu) = \bar{S}\#\mu$, therefore $\int \|x - \bar{S}(x)\|^2 d\mu(x) \geq W_2^2(\mu, G(\mu))$. Finally, since $S_\mu^{\nu_i}\#\mu \leq_c \nu_i$, we have that $\int \|\bar{S}(x) - S_\mu^{\nu_i}(x)\|^2 d\mu(x) \geq V(G(\mu)|\nu_i)$. This , recalling that $V(\mu|\nu_i) = W_2^2(\mu, S_\mu^{\nu_i}\#\mu)$, yields

$$\sum_{i=1}^n \lambda_i V(\mu|\nu_i) \geq \sum_{i=1}^n \lambda_i V(G(\mu)|\nu_i) + W_2^2(\mu, G(\mu)). \tag{19}$$

Therefore, if $\mu$ is a weak barycenter, we readily get that $\mu = G(\mu)$. $\square$

*Proof of Proposition 5.* As in the proof of Theorem 4, we denote $S_k^i$ the optimal barycentric projection associated to $\pi^{k,i} \in \Pi(\mu_k, \nu_i)$. First, we easily have that $\mu_{k+1} \in \mathcal{P}_2(\mathbb{R}^d)$, indeed by Jensen's inequality

$$\int \|x\|^2 d\mu_{k+1}(x) = \iint \|\sum_{i=1}^n \lambda_i S_k^i(x)\|^2 d\mu_k(x) \leq \sum_{i=1}^n \lambda_i \int \|y\|^2 d\nu_i(y) < \infty.$$

Then $(\mu_k)_k$ is tight, with uniformly integrable 2-moments by Lemma 5. Therefore $(\mu_k)_k$ admits a convergent subsequence in $W_2$. Let $\tilde{\mu}$ be a weak limit of a subsequence $(\mu_{k_j})_j$, then we have $W_2(\mu_{k_j}, \tilde{\mu}) \xrightarrow[j\to\infty]{} 0$. By continuity of $G$ in Theorem 4, we get $W_2(\mu_{k_j+1}, G(\tilde{\mu})) \xrightarrow[j\to\infty]{} 0$. Moreover, by Theorem 6 we have for $F(\mu) := \sum_{i=1}^n \lambda_i V(\mu|\nu_i)$ that $F(\mu_{k_j}) \to F(\tilde{\mu})$ and $F(\mu_{k_j+1}) \to F(G(\tilde{\mu}))$ as $j \to \infty$ . Let us prove that these two limits coincide. From (19), we have

$$F(\mu_{k_j}) \geq \sum_{i=1}^n \lambda_i V(G(\mu_{k_j})|\nu_i) = \sum_{i=1}^n \lambda_i V(\mu_{k_j+1}|\nu_i) = F(\mu_{k_j+1}).$$

Iterating this inequality leads to $F(\mu_{k_j}) \geq F(\mu_{k_j+1}) \geq F(\mu_{k_{j+1}})$ which yields $F(\tilde{\mu}) = F(G(\tilde{\mu}))$ and then $\tilde{\mu} = G(\tilde{\mu})$, using inequality (19). Thus $(\mu_{k_j})_j$ converges w.r.t. $W_2$ to a probability distribution $\tilde{\mu}$ which is a fixed point of $G$.

$\square$

*Proof of Lemma 4.* The proof is similar to that of [35, Lemma 3.8]. For the sake of clarity, we rewrite it in our setting. We assume that $x = \int S_{\bar{\mu}}^{\nu}(x)\mathrm{d}\mathbb{Q}(\nu)$, $\bar{\mu}(x)$-a.s. is not true, then

$$0 < \int \|x - \int S_{\bar{\mu}}^{\nu}(x)\mathrm{d}\mathbb{Q}(\nu)\|^2 \mathrm{d}\bar{\mu}(x)$$

$$= \int \|x\|^2\mathrm{d}\bar{\mu}(x) - 2\iint \langle x, S_{\bar{\mu}}^{\nu}(x)\rangle \mathrm{d}\mathbb{Q}(\nu)\mathrm{d}\bar{\mu}(x) + \int \|\int S_{\bar{\mu}}^{\nu}(x)\mathrm{d}\mathbb{Q}(\nu)\|^2\mathrm{d}\bar{\mu}(x).$$

Moreover, $S_{\bar{\mu}}^{\mu}\#\bar{\mu} \leq_c \mu$, therefore by Theorem 1.4 in [8], we get

$$\int V\left(\left[\int S_{\bar{\mu}}^{\nu}\mathrm{d}\mathbb{Q}(\nu)\right]\#\bar{\mu}|\mu\right)\mathrm{d}\mathbb{Q}(\mu) \leq \int \|\int S_{\bar{\mu}}^{\nu}\mathrm{d}\mathbb{Q}(\nu) - S_{\bar{\mu}}^{\mu}\|_{\mathbb{L}^2(\bar{\mu})}^2 \mathrm{d}\mathbb{Q}(\mu)$$

$$= \iint \|S_{\bar{\mu}}^{\nu}(x)\|^2\mathrm{d}\bar{\mu}(x)\mathrm{d}\mathbb{Q}(\nu) - \int \|\int S_{\bar{\mu}}^{\nu}\mathrm{d}\mathbb{Q}(\nu)\|^2\mathrm{d}\bar{\mu}(x).$$

Finally, noticing that $\iint \|x - S_{\bar{\mu}}^{\nu}(x)\|^2\mathrm{d}\bar{\mu}(x)\mathrm{d}\mathbb{Q}(\nu) = \int V(\bar{\mu}|\nu)\mathrm{d}\mathbb{Q}(\nu)$, we hence get

$$\int V\left(\left[\int S_{\bar{\mu}}^{\nu}\mathrm{d}\mathbb{Q}(\nu)\right]\#\bar{\mu}|\mu\right)\mathrm{d}\mathbb{Q}(\mu) < \int V(\bar{\mu}|\nu)\mathrm{d}\mathbb{Q}(\nu),$$

which is in contradiction with $\bar{\mu}$ weak barycenter of $\mathbb{Q}$. □

In order to study the convergence of the iterative scheme in (10), we define the following objects:

$$L(\mu) := \frac{1}{2}\int V(\mu|\nu)\mathrm{d}\mathbb{Q}(\nu) \tag{20}$$

$$H(\mu)(x) := -\int (S_{\mu}^{\nu} - \mathrm{id})\mathrm{d}\mathbb{Q}(\nu)(x) \quad x \in \mathbb{R}^d. \tag{21}$$

Moreover, we denote by $\{\mathcal{F}_k\}_k$ the filtration of the i.i.d. sample $\nu^k \sim \mathbb{Q}$, namely $\mathcal{F}_{-1}$ is the trivial sigma-algebra and $\mathcal{F}_{k+1}$ is the sigma-algebra generated by $\nu^0, \ldots, \nu^k$ and therefore $\mu_k$ in (10) is $\mathcal{F}_k$-measurable.

The next two Propositions are needed to prove Theorem 5.

**Proposition 6.** *The functions* $\mu \in \mathcal{P}_2(\mathbb{R}^d) \mapsto \|H(\mu)\|_{\mathbb{L}^2(\mu)}^2$ *and* $\mu \in \mathcal{P}_2(\mathbb{R}^d) \mapsto L(\mu)$ *are continuous w.r.t* $W_2$.

*Proof.* Let us first assume that $(\rho_m)_m, \rho$ in $\mathcal{P}_2(\mathbb{R}^d)$ are such that $W_2(\rho_m, \rho) \to 0$. We want to prove that

$$\|H(\rho_m)\|_{\mathbb{L}^2(\rho_m)}^2 \to \|H(\rho)\|_{\mathbb{L}^2(\rho)}^2 \tag{22}$$

when $m \to \infty$. Consider the probability space $(\Omega, \mathcal{F}, \mathbb{P})$ and r.v.'s $(X_m)_m$ and $X$ constructed in Lemma 6.ii), and recall that, for each $\nu \in \mathcal{P}_2(\mathbb{R}^d)$, the r.v.'s $(X_m, S_{\rho_m}^{\nu}(X_m))$ have law $(\mathrm{id}, S_{\rho_m}^{\nu})\#\rho_m$ for each $m$ and converge in $\mathbb{L}^2(\Omega, \mathcal{F}, \mathbb{P})$ to the r.v. $(X, S_{\rho}^{\nu}(X))$, which has the law $(\mathrm{id}, S_{\rho}^{\nu})\#\rho$. We next extend this construction in order to suitably randomise $\nu$. More precisely, we enlarge the probability space $(\Omega, \mathcal{F}, \mathbb{P})$ to the product space $(\bar{\Omega}, \bar{\mathcal{F}}, \bar{\mathbb{P}}) = (\Omega \times \mathcal{P}_2(\mathbb{R}^d), \mathcal{F} \otimes \mathcal{B}(\mathcal{P}_2(\mathbb{R}^d)), \mathbb{P} \otimes \mathbb{Q})$, that is, we add an independent random variable, called $\boldsymbol{\nu}$, taking values in $\mathcal{P}_2(\mathbb{R}^d)$ and which has distribution $\mathbb{Q}$.

Thanks to the measurability of the mappings $(x, \nu) \mapsto S_{\rho_m}^{\nu}$ and $(x, \nu) \mapsto S_{\rho}^{\nu}$ proven in Lemma 3, by replacing $\nu$ by $\boldsymbol{\nu}$ in the previous objects we obtain random vectors $(X_m, S_{\rho_m}^{\boldsymbol{\nu}}(X_m))$ and $(X, S_{\rho}^{\boldsymbol{\nu}}(X))$ defined in $(\bar{\Omega}, \bar{\mathcal{F}}, \bar{\mathbb{P}})$ which have, conditionally on $\{\boldsymbol{\nu} = \nu\}$, the laws $(\mathrm{id}, S_{\rho_m}^{\nu})\#\rho_m$ and $(\mathrm{id}, S_{\rho}^{\nu})\#\rho$ respectively. Moreover, $\boldsymbol{\nu}$ is independent of the r.v. $X, X_1, \ldots X_m$ under $\bar{\mathbb{P}}$.

Now, by conditioning on $\{\boldsymbol{\nu} = \nu\}$, using the convergence result in Lemma 6.ii) and the dominated convergence Theorem, we can easily check that $((X_m, S_{\rho_m}^{\boldsymbol{\nu}}(X_m)))_m$ converges to $(X, S_{\rho}^{\boldsymbol{\nu}}(X))$ in $\bar{\mathbb{P}}$−probability. Furthermore, one can integrate w.r.t. $\mathbb{Q}$ the bound (18) obtained for fixed $\nu$ in the proof of Lemma 5 and, denoting $\bar{\mathbb{E}}$ the expectation with respect to $\bar{\mathbb{P}}$, deduce that

$$\sup_m \bar{\mathbb{E}}\left(\|S_{\rho_m}^{\boldsymbol{\nu}}(X_m)\|^2\mathbf{1}_{\{\|S_{\rho_m}^{\boldsymbol{\nu}}(X_m)\|^2 \geq M\}}\right) \leq \int\int_{\{\|x\|^2 \geq K\}} \|x\|^2\mathrm{d}\nu(x)\mathbb{Q}(\mathrm{d}\nu)$$

$$+ \frac{K}{M}\int\int \|x\|^2\mathrm{d}\nu(x)\mathbb{Q}(\mathrm{d}\nu),$$

for each $M, K \geq 0$, where the r.h.s. is finite since $\mathbb{Q} \in \mathcal{P}_2(\mathcal{P}_2(\mathbb{R}^d))$. It follows that the sequence $((X_m, S_{\rho_m}^{\boldsymbol{\nu}}(X_m)))_m$ has uniformly integrable second moments, and therefore converges also in $L^2(\bar{\Omega}, \bar{\mathcal{F}}, \bar{\mathbb{P}})$ to $(X, S_\rho^{\boldsymbol{\nu}}(X))$, thanks to the Vitali convergence theorem. In particular, as $m$ tends to infty, we have

$$\int V(\mu_m|\nu)\mathrm{d}\mathbb{Q}(\nu) = \mathbb{E}\|X_m - S_{\rho_m}^{\boldsymbol{\nu}}(X_m)\|^2 \to \mathbb{E}\|X - S_\rho^{\boldsymbol{\nu}}(X)\|^2 = \int V(\mu|\nu)\mathrm{d}\mathbb{Q}(\nu),$$

which proves the continuity of the function $\mu \in \mathcal{P}_2(\mathbb{R}^d) \mapsto L(\mu)$.

We observe now that $\bar{\mathbb{E}}\left(S_{\rho_m}^{\boldsymbol{\nu}}(X_m)|X_m\right) = \int S_{\rho_m}^\nu(X_m)\mathrm{d}\mathbb{Q}(\nu)$ and $\bar{\mathbb{E}}\left(S_\rho^{\boldsymbol{\nu}}(X)|X\right) = \int S_\rho^\nu(X)\mathrm{d}\mathbb{Q}(\nu)$, $\bar{\mathbb{P}}-$ a.s., Moreover, if $\mathcal{F}_\infty$ denotes the $\sigma$-algebra generated by $(X_1, X_2, \ldots)$, one has $\bar{\mathbb{E}}\left(S_{\rho_m}^{\boldsymbol{\nu}}(X_m)|\mathcal{F}_\infty\right) = \bar{\mathbb{E}}\left(S_{\rho_m}^{\boldsymbol{\nu}}(X_m)|X_m\right)$ and $\bar{\mathbb{E}}\left(S_\rho^{\boldsymbol{\nu}}(X)|\mathcal{F}_\infty\right) = \bar{\mathbb{E}}\left(S_\rho^{\boldsymbol{\nu}}(X)|X\right)$. Using the continuity in $L^2(\bar{\Omega}, \bar{\mathcal{F}}, \bar{\mathbb{P}})$ of the conditional expectation with respect to $\mathcal{F}_\infty$, we deduce that

$$X_m - \bar{\mathbb{E}}\left(S_{\rho_m}^{\boldsymbol{\nu}}(X_m)|X_m\right) \to X - \bar{\mathbb{E}}\left(S_\rho^{\boldsymbol{\nu}}(X)|X\right) \tag{23}$$

in $L^2(\bar{\Omega}, \bar{\mathcal{F}}, \bar{\mathbb{P}})$. We conclude that $\bar{\mathbb{E}}\|X_m - \bar{\mathbb{E}}(S_{\rho_m}^{\boldsymbol{\nu}}(X_m)|X_m)\|^2 \to \bar{\mathbb{E}}\|X - \bar{\mathbb{E}}S_\rho^{\boldsymbol{\nu}}(X)|X)\|^2$ as $m \to \infty$, which is exactly the required convergence (22).

$\square$

**Proposition 7.** *For the sequence $(\mu_k)_k$ defined in (10), we have*

$$\mathbb{E}(L(\mu_{k+1}) - L(\mu_k)|\mathcal{F}_k) \leq \gamma_k^2 L(\mu_k) - \gamma_k\|H(\mu_k)\|_{\mathbb{L}^2(\mu_k)}^2. \tag{24}$$

*Proof.* The arguments are similar to the ones used for the population Wasserstein barycenter iterative scheme in the proof of Proposition 3.2 in [10]. Let us set them for the present problem. Let $\nu \in \mathrm{supp}(Q)$, then $([(1 - \gamma_k)\mathrm{id} + \gamma_k S_{\mu_k}^{\nu^k}], S_{\mu_k}^\nu]\#\mu_k$ belongs to $\Pi(\mu_{k+1}, S_{\mu_k}^\nu\#\mu_k)$. Therefore we have

$$\begin{aligned}
V(\mu_{k+1}|\nu) &\leq W_2^2(\mu_{k+1}, S_{\mu_k}^\nu\#\mu_k) \quad \text{since } S_{\mu_k}^\nu\#\mu_k \leq_c \nu \\
&\leq \int \|(1 - \gamma_k)x + \gamma_k S_{\mu_k}^{\nu^k}(x) - S_{\mu_k}^\nu(x)\|^2\mathrm{d}\mu_k(x) \\
&= \int \|x - S_{\mu_k}^\nu(x)\|^2\mathrm{d}\mu_k(x) - 2\gamma_k \int \langle x - S_{\mu_k}^\nu(x), x - S_{\mu_k}^{\nu^k}(x)\rangle\mathrm{d}\mu_k(x) \\
&\quad + \gamma_k^2 \int \|x - S_{\mu_k}^{\nu^k}(x)\|^2\mathrm{d}\mu_k(x) \\
&= V(\mu_k|\nu) + \gamma_k^2 V(\mu_k|\nu^k) - 2\gamma_k \int \langle x - S_{\mu_k}^\nu(x), x - S_{\mu_k}^{\nu^k}(x)\rangle\mathrm{d}\mu_k(x).
\end{aligned}$$

Integrating with respect to $\nu$, and divided by 2 we get

$$L(\mu_{k+1}) \leq L(\mu_k) + \frac{\gamma_k^2}{2}V(\mu_k|\nu^k) - \gamma_k \int \langle H(\mu_k)(x), x - S_{\mu_k}^{\nu^k}(x)\rangle\mathrm{d}\mu_k(x).$$

We can then take the conditional expectation with respect to the filtration $\mathcal{F}_k$, knowing that $\mu_k$ is $\mathcal{F}_k$-measurable and that $\nu^k$ is independently sampled from $\mathcal{F}_k$, we have

$$\begin{aligned}
\mathbb{E}(L(\mu_{k+1})|\mathcal{F}_k) &\leq L(\mu_k) + \frac{\gamma_k^2}{2}\int V(\mu_k|\nu)\mathrm{d}\mathbb{Q}(\nu) - \gamma_k \int \langle H(\mu_k)(x), \int x - S_{\mu_k}^\nu(x)\mathrm{d}\mathbb{Q}(\nu)\rangle\mathrm{d}\mu_k(x) \\
&= L(\mu_k) + \gamma_k^2 L(\mu_k) - \gamma_k \int \langle H(\mu_k)(x), H(\mu_k)(x)\rangle\mathrm{d}\mu_k(x) \\
&= (1 + \gamma_k^2)L(\mu_k) - \gamma_k\|H(\mu_k)\|_{\mu_k}^2.
\end{aligned}$$

$\square$

*Proof of Theorem 5.* We will proceed in a similar way as in the proof of Theorem 1.4 in [10]. Let us first note that the set $K_\mathbb{Q}$ is compact in $\mathcal{P}_2(\mathbb{R}^d)$ w.r.t. $W_2$ (see [42]). Moreover, the sequence $(\mu_k)_k$ is a.s. included in this compact set, as can be seen by induction using the facts that the function $|\cdot|^{2+\epsilon}$

is convex and that $S^{\nu_k}_{\mu_k} \# \mu_k \leq_c \nu_k$, with $\nu^k \sim \mathbb{Q}$. Now let $\bar{\mu}$ be a weak population barycenter, i.e. $\bar{\mu}$ minimises $L$ defined in (20), and write $\bar{L} := L(\bar{\mu})$. We introduce the sequences

$$h_k := L(\mu_k) - \bar{L} \geq 0 \quad \text{and} \quad \alpha_k := \prod_{i=1}^{k-1} \frac{1}{1 + \gamma_k^2}.$$

From condition (11), the sequence $(\alpha_k)_k$ converges to some $\alpha_\infty > 0$. By Proposition 7, we have

$$\mathbb{E}(h_{k+1} - (1 + \gamma_k^2)h_k | \mathcal{F}_k) \leq \gamma_k^2 \bar{L} - \gamma_k \|H(\mu_k)\|_{\mu_k}^2 \leq \gamma_k^2 \bar{L}$$
$$\Rightarrow \mathbb{E}(\alpha_{k+1}h_{k+1} - \alpha_k h_k | \mathcal{F}_k) \leq \alpha_{k+1}\gamma_k^2 \bar{L} \quad \text{upon multiplying by } \alpha_{k+1}. \tag{25}$$

Defining now

$$\delta_k := \begin{cases} 1 & \text{if } \mathbb{E}(\alpha_{k+1}h_{k+1} - \alpha_k h_k | \mathcal{F}_k) > 0 \\ 0 & \text{otherwise}, \end{cases}$$

we deduce that

$$\sum_{k=1}^\infty \mathbb{E}(\delta_k(\alpha_{k+1}h_{k+1} - \alpha_k h_k)) = \sum_{k=1}^\infty \mathbb{E}(\delta_k \mathbb{E}(\alpha_{k+1}h_{k+1} - \alpha_k h_k | \mathcal{F}_k))$$

$$\leq L(\bar{\mu}) \sum_{k=1}^\infty \alpha_{k+1}\gamma_k^2 \leq L(\bar{\mu}) \sum_{k=1}^\infty \gamma_k^2 < \infty.$$

Since $h_k \alpha_k \geq 0$, by the quasi-martingale convergence theorem $(h_k \alpha_k)_k$ converges almost surely. Since $(\alpha_k)_k$ converges to $\alpha_\infty > 0$, then $(h_k)_k$ also converges almost surely to some $h_\infty \geq 0$. Taking expectations in Eq. (25) and summing we get

$$\mathbb{E}(\alpha_{k+1}h_{k+1}) \leq \mathbb{E}(\alpha_0 h_0) + \bar{L} \sum_{m=1}^k \alpha_{m+1}\gamma_m^2 \leq C.$$

Fatou's Lemma yields $\mathbb{E}(\alpha_\infty h_\infty) < \infty$, and since $\alpha_\infty < \infty$, we deduce that $h_\infty$ is almost surely finite. Our goal now is to show that $h_\infty = 0$. From (25) we deduce as before that

$$\mathbb{E}(\alpha_{k+1}h_{k+1}) - \mathbb{E}(\alpha_0 h_0) \leq \bar{L} \sum_{m=1}^k \alpha_{m+1}\gamma_m^2 - \sum_{m=1}^k \alpha_{m+1}\gamma_m \mathbb{E}\left( \|H(\mu_m)\|_{\mathbb{L}_2(\mu_m)}^2 \right).$$

Taking liminf, using Fatou on the l.h.s. and monotone convergence on the r.h.s. we obtain that

$$-\infty < \mathbb{E}(\alpha_\infty h_\infty) - \mathbb{E}(\alpha_0 h_0) \leq C - \mathbb{E}\left( \sum_{m=1}^\infty \alpha_{m+1}\gamma_m \|H(\mu_m)\|_{\mathbb{L}_2(\mu_m)}^2 \right)$$

hence, in particular,

$$\sum_{k=1}^\infty \gamma_k \|H(\mu_k)\|_{\mathbb{L}_2(\mu_k)}^2 < +\infty \quad \text{a.s.} \tag{26}$$

Note that the conditions in Eqs. (26) and (11) imply that

$$\liminf_{t\to\infty} \|H(\mu_k)\|_{\mathbb{L}_2(\mu_k)}^2 = 0, \text{ a.s.} \tag{27}$$

Observe also that, by the compactness of $K_\mathbb{Q}$ and the continuity of $L$ in Proposition 6, the set $\{\rho : L(\rho) \geq \bar{L} + \delta\} \cap K_\mathbb{Q}$ is $W_2$-compact. Therefore, the function $\rho \mapsto \|H(\rho)\|_{\mathbb{L}^2(\rho)}^2$, also $W_2$-continuous by Proposition 6, attains its minimum on that set. That minimum cannot be zero, as otherwise we would have obtained a fixed-point that is not a weak barycenter, contradicting our hypothesis. It follows that

$$\forall \delta > 0, \inf_{\{\rho : L(\rho) \geq \bar{L} + \delta\} \cap K_\mathbb{Q}} \|H(\rho)\|_{\mathbb{L}^2(\rho)}^2 > 0. \tag{28}$$

Since $\{\mu_k\}_k \subset K_\mathbb{Q}$ a.s., we deduce from the previous result the a.s. inclusions of events:

$$\{h_\infty \geq 2\delta\} \subset \{\mu_t \in \{\rho : L(\rho) \geq L(\bar{\mu}) + \delta\} \cap K_\mathbb{Q} \; \forall t \text{ large enough}\}$$

$$\subset \bigcup_{\ell \in \mathbb{N}} \left\{ \|H(\mu_t)\|_{\mathbb{L}^2(\mu_t)}^2 > 1/\ell : \forall t \text{ large enough} \right\} \subset \left\{ \liminf_{t\to\infty} \|H(\mu_t)\|_{\mathbb{L}^2(\mu_t)}^2 > 0 \right\},$$

where Eq. (28) was used to obtain the second inclusion. It follows using Eq. (27) that $\mathbb{P}(h_\infty \geq 2\delta) = 0$ for every $\varepsilon > 0$, hence $h_\infty = 0$ a.s. as claimed. In other words, $L(\mu_k) \to \bar{L}$ a.s. as $k \to \infty$.

We already established that $\{\mu_k\}_k \subset K_{\mathbb{Q}}$, hence the sequence is relatively compact. We finally conclude that the limit $\hat{\mu}$ of any convergent subsequence $\{\mu_{k_j}\}_{k_j}$ satisfies $L(\hat{\mu}) = \lim_j L(\mu_{k_j}) = \bar{L}$, whence, it is a weak barycenter.

**Remark 2.** *Assumption (A) can be replaced by the following more general condition:*

*(A')*     $\mathbb{Q}$ *gives full measure to a* $W_2$-*compact set* $K_{\mathbb{Q}}$ *which is* "*weakly geodesically closed*", *in the sense that for any* $\mu, \nu \in K_{\mathbb{Q}}$ *and* $t \in [0, 1]$, $((1-t)\mathrm{id} + tS_\mu^\nu)\#\mu \in K_{\mathbb{Q}}$.

$\square$

# E    Numerical results

## E.1    Proximal algorithm for the computation of the OWT plan

This section is dedicated to the resolution of the OWT problem. Let $\mu = \sum_{i=1}^{r} a_i \delta_{x_i}$ and $\nu = \sum_{j=1}^{m} b_i \delta_{y_j}$, be two discrete measures, the OWT problem boils down to solving

$$\min_{\pi \in \mathbb{R}^{r \times m}} \underbrace{\sum_{i=1}^{r} a_i \|x_i - \left(\frac{\pi \mathbf{y}}{\mathbf{a}}\right)_i\|^2}_{f(\pi)} + \underbrace{1_{\Pi(\mu,\nu)}(\pi)}_{g(\pi)}, \tag{29}$$

where $1_C$ is the indicator function of the set $C$ i.e.

$$1_C(\pi) = \begin{cases} \pi & \text{if } \pi \in C \\ \infty & \text{otherwise.} \end{cases}$$

The proximal algorithm to solve Eq. (29) then reads:

$$\pi^{\ell+1} = \mathrm{prox}_{\theta^\ell g}(\pi^\ell - \theta^\ell \nabla f(\pi^\ell)). \tag{30}$$

As $\Pi(\mu, \nu)$ is a closed non-empty convex set, the proximal operator of $g$ reduces to the Euclidean projection onto $\Pi(\mu, \nu)$:

$$\mathrm{proj}_{\Pi(\mu,\nu)}(P) = \arg\min_{\pi \in \mathbb{R}^{r \times m}} \|P - \pi\|^2 = \arg\min_{\pi \in \mathbb{R}^{r \times m}} \langle \pi, -P \rangle + \frac{1}{2}\|\pi\|^2$$

where $\|\cdot\|$ is the Frobenius norm. This projection problem can be solved by Dykstra's algorithm with alternate Bregman projections [20] or by stochastic dual approaches of OT regularised by an $\mathbb{L}_2$ norm [36]. This method is summarised in Algorithm 3. In particular, we used an accelerated version of Eq. (30) via FISTA [12] (with $\omega_\ell \in [0, 1)$ an extrapolation parameter and $\theta_\ell$ the usual stepsize chosen by a line search) in order to compute the optimal plan $\pi_\mu^\nu$ in the weak transport problem.

The optimal barycentric projection is then given by $S_\mu^\nu = \frac{\pi_\mu^\nu \mathbf{y}}{\mathbf{a}}$. We initialised the algorithm with a random matrix whose elements sum to 1. Observe that, from Algorithm 1, the $K$ optimal barycentric projection computations can be parallelised for each step $n$.

---

**Algorithm 3:** Computation of the optimal weak plan

---

**Output:** $\pi_\mu^\nu$;
**Input:** $\mu = \sum_{i=1}^{r} a_i \delta_{x_i}$ and $\nu = \sum_{j=1}^{m} b_j \delta_{y_j}$;
Initialise $\pi_0$ random matrix;
**while** *not converge* **do**
     $P_{\ell+1} := \pi_\ell + \omega_\ell(\pi_\ell - \pi_{\ell-1})$;
     $\pi_{\ell+1} := \mathrm{proj}_{\Pi(\mu,\nu)}(P_{\ell+1} - \theta_\ell \nabla f(P_{\ell+1}))$;
**end**

---

With respect to the efficiency of this algorithm, Figure 7 shows a comparison of different settings for Eq. (30) in order to compute an optimal weak transport plan. For that purpose, we considered

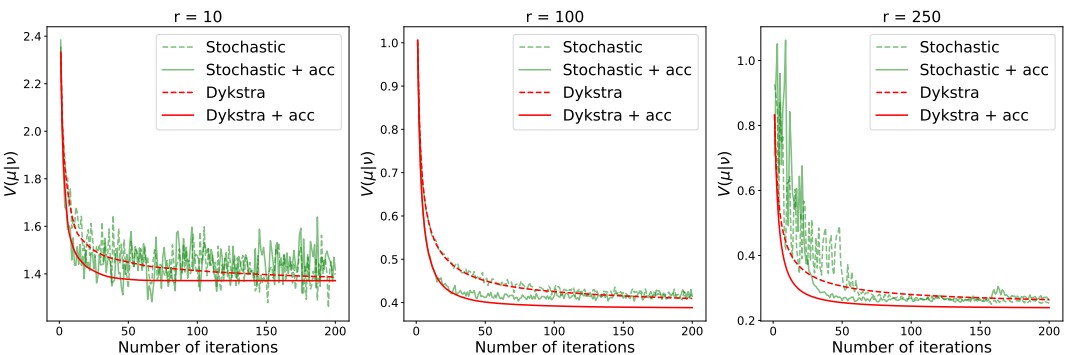

Figure 7: Convergence of the algorithm (3) in several settings for measures $\mu$ and $\nu$ supported on $r = m$ points.

two discrete distributions $\mu$ and $\nu$ each constructed from $r = m = 10, 100$ and $250$ samples of two dimensional Gaussian measures. We illustrate the convergence for both the standard and accelerated versions of the proximal algorithm, as well as for the projection into $\Pi(\mu, \nu)$ via Dykstra's algorithm or the stochastic dual approach. As expected, the accelerated version of Eq. (30) converges faster than the classical proximal algorithm, and the projection step in more stable with Dykstra's algorithm. Moreover, the smaller the number of support points, the faster the convergence. We have also noted that the random initialisation does not affect the convergence towards the minimiser of Eq. (29).

### E.2 Additional experiments

**Gaussian distributions** As in Section 5 of [5], we computed a weak barycenter between two 2D centered ellipses $E(\Sigma_i) = \{s \in \mathbb{R}^2 : x^t \Sigma_i^{-1} x = 1\}$ with covariances matrices

$$\Sigma_1 = \begin{pmatrix} 2 & 0 \\ 0 & 1 \end{pmatrix} \quad \text{and} \quad \Sigma_2 = \begin{pmatrix} 1 & 0 \\ 0 & 2 \end{pmatrix},$$

by considering 300 random observations foe each ellipse. We then executed the iterations of Algorithm 1 until the difference of the objective function (i.e., the sum in Eq. (5)) between two successive iterations was smaller than $1e - 5$. This occurred at the 8th iteration, and the resulting weak barycenter was a circle within both ellipses. As we have access to the value of the weak barycenter problem (see Eq. (7)), we also compared the value of the objective function at the 8th iteration (that is $3.62e - 4$) to $\frac{1}{2} \sum_{i=1}^{2} \|\mathbb{E}(Y_i)\|^2 - \|\frac{1}{2} \sum_{i=1}^{2} \mathbb{E}(Y_i)\|^2$, with a plug-in estimator for $\mathbb{E}(Y_i)$. The approximated objective was equal to $3.21e - 4$, therefore, Algorithm 1 gave a satisfactory optimised weak barycenter.

**Ellipse distributions** ($r = 100 \ \& \ K = 15$). We considered ellipse distributions with random center in $(-5, 5)$, random semi-major and semi-minor axes in $(6, 14)$. The results are presented in Fig. 8, where the same conclusions as in the Gaussian examples hold.

**Pair-of-ellipses** ($r \in (200, 300) \ \& \ K = 10$). In the same fashion, we considered distributions supported on two ellipses with random centers in $(-5, 5)$, random semi-major and semi-minor axes in $(1, 7)$ and $(7, 13)$ respectively. Fig. 9 shows the distributions (left) as well as the OT and OWT barycenters (right) computed from random samples of the distributions. Observe that, once again, the weak barycenter better preserved the structure of the distributions when computing Algorithm 2.

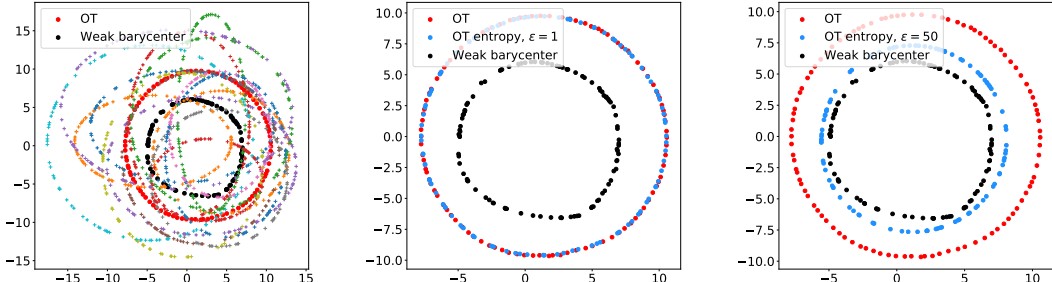

Figure 8: (left) Ellipse distributions and their OWT (black) and OT (red) barycenters computed with Algorithm 2. (center & right) Illustration of the weak (black), OT (red) and OT Sinkhorn (blue) barycenters for different values of $\varepsilon = 1, 50$.

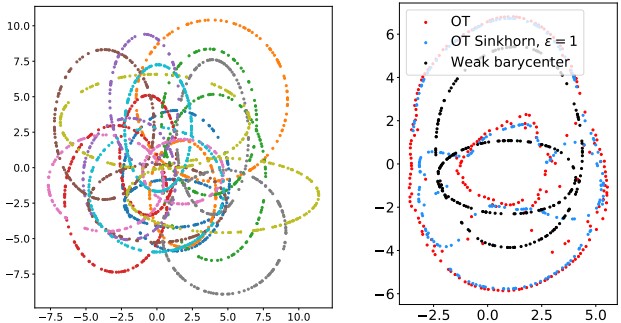

Figure 9: (left) Distributions supported on a pair-of-squares. (right) OWT (black), OT (red) and OT Sinkhorn for $\varepsilon = 1$ (blue) barycenters computed with Algorithm 2.