# OpenReview forum: "A novel notion of barycenter for probability distributions based on optimal weak mass transport"
_NeurIPS.cc/2021/Conference — NeurIPS 2021 Poster_

### Official Review · Reviewer_KZXz · 2021-07-10

**Rating:** 6
**Confidence:** 3

**Summary:**

This paper studies barycenters of probability distributions using the weak transport distance, and establishes a number of results which mirror the theory of standard optimal transport barycenters that has been established over the past 10 years. In particular, they investigate existence and uniqueness, the first order optimality criterion, as well as iterative methods to compute weak barycenters. They propose a proximal sub-routine for this iterative method, and evaluate the performance of weak barycenters on a number of examples, comparing with standard barycenters and Sinkhorn barycenters.

**Limitations And Societal Impact:**

The authors adequately discuss limitations (including references for the standard barycenter version of their proofs plus valid directions for extensions) and societal impact.

**Main Review:**

The paper is the first to study weak optimal transport barycenters, and establishes a number of desirable results for this new notion. In particular, the authors establish existence (in both finitely supported and non-finitely supported Q cases) and some optimality properties of weak barycenters, such as closure under the convex order. They also give a latent variable characterization of the weak barycenter and study first order methods for computing the weak barycenter. Finally, they provide convincing and clear empirical results that there is significant value in this weak barycenter notion.

Overall, I believe this is a quality, well-written submission. Although the technical results are largely based off porting over proofs from the standard case, I think the convincing empirics demonstrate that this is certainly a valuable notion, and the breadth of technical results is of substantial value for the literature.

**Time Spent Reviewing:**

2

---

> ### Author Response · Authors · 2021-08-10
> **We elaborate on the strenght of our paper in terms of intermediate results.**
>
> We would like to thank you for your positive assessment and pointing out the theoretical work that we have provided in this paper. In particular, we believe that the intermediate lemmas proved in this paper, that are completely novel, can be very relevant for the optimal weak transport community, namely Lemma 3, Lemma 5 and Lemma 6, on measurability and continuity properties of the optimal barycentric projection in the optimal weak transport sense. These properties are crucial and fundamental in order to build sensitive and solid results on optimal weak transport objects.

---

> > ### Comment · Reviewer_KZXz · 2021-08-25
> > **Thanks**
> >
> > Thanks for your reply!

---

### Official Review · Reviewer_kNgK · 2021-07-13

**Rating:** 7
**Confidence:** 4

**Summary:**

This paper introduces and study Fréchet means (aka barycenter) problem for optimal weak transport (OWT), a variant of the standard optimal transport problem which presents in particular the benefits of guaranteeing the existence of optimal transport *map* without needing regularity assumptions on the measures. They prove theoretical properties of these objects (existence, characterization) and, using a fixed-point equation that any (OWT) barycenter must satisfy, they design an interative algorithm that is guaranteed to converge toward such a fixed-point (which, however, may not be a OWT barycenter).

**Limitations And Societal Impact:**

The paper mentions some of its limitations/questions left. I did not identify negative social impact that may be specific to this work.

**Main Review:**

Quality:
The paper introduces an interesting optimization problem and studies it properly. The theoretical content is solid, it comes with algorithms + guarantees, and numerical experiments - though not ground-breaking as restricted to dimension 2 - support the approach in an informative way.

Clarity:
The paper is well-written and pleasant to read.

Originality / Significance:
OWT is a fairly new topic in the Optimal Transport field and this paper addresses a natural question when new OT models are proposed. As such, it may open the way for interesting theoretical and applied use of O(W)T in machine learning and other fields. Important questions remain open, in particular the existence (and perhaps the estimation) of a ''maximal'' OWT barycenter when $d \geq 2$. In particular, the discrepancy between the existence of a "trivial" OWT barycenter (Lemma 1) and the (interesting) output of the proposed Algorithms suggest that some gap should be filled in the understanding of fixed-points of the map $\mu \mapsto G(\mu)$.


Minor comments and suggestions:
- As (for once!) we exactly know the objective value of a (true) OWT barycenter (Eq 7), it may be enlightening to show how does the output of the proposed Algorithms perform empirically in terms of energy. Perhaps does it avoid spurious local minima (if non-adversarialy initialized) and actually converges toward a true OWT barycenter "in practice".
- Figure 9 (Pair of ellipses) may be worth including in the main paper as it feels quite convincing.
- It may be nice to present an experiment (perhaps the Pair of ellipses?) in the formalism of Theorem 3, showing that (hopefully) one retrieves (some valid) $X$ given the $Y_i$.


**Time Spent Reviewing:**

5

---

> ### Author Response · Authors · 2021-08-10
> **We discuss the "maximal" weak barycenter, and experiments that could best benefit from our proposal.**
>
> We thank the reviewer for their positive comments and feedback on our work. We next elaborate on your comments.
>
> **(i)** As pointed out in the paper, the natural question of the existence of a "maximal" weak barycenter within the set of all solutions is very difficult to answer. However, a supportive intuition in terms of robustness is that a maximal weak barycenter would be one that includes the most possible points of all classes (or distributions) in its support (all this, after re-centering) and leaves out only "outliers".  A non-maximal weak barycenter is then more conservative, meaning that it counts on fewer points and leaves out more possible outliers.
>
> **(ii)** Thank you for suggesting the study of the objective function (as the true objective is known). We have conducted a simple and informative analysis, described in the response to Reviewer Ee1U item (iii).
>
> **(iii)** We agree that the Pair-of-ellipses experiments would improve the understanding of our contribution and might convince a wider readership. We will accommodate this example in the main document.
>
> **(iv)** We agree that an illustration of that sort would be of great interest. We intend to produce an experiment of the latent variable formulation in order to provide more insight in that regard.

---

> > ### Comment · Reviewer_kNgK · 2021-08-25
> > **Thanks**
> >
> > Thank you for taking time answering my comments.
> > Item *(iii)* in your answer to Ee1U is interesting and satisfactory to me. I think generalizing this sanity-check (investigating if one empirically obtains a near-optimal weak-OTB) in the paper would support the proposed algorithm.

---

> > > ### Author Response · Authors · 2021-08-28
> > > **Reply**
> > >
> > > We agree that comparing the value of the computed weak barycenter against the approximated optimal value (using the plug-in estimator) is beneficial, in particular towards supporting a wider use of the proposed algorithm. We thank the reviewer for suggesting this practice. In the revised version of the paper, we will recommend its systematic use, and complement our experiments with similar tests as described in the second part of point (iii) of the answer to reviewer Ee1U.

---

### Official Review · Reviewer_qhuc · 2021-07-17

**Rating:** 7
**Confidence:** 4

**Summary:**

The present manuscript introduces the natural extension of the concept of the Wasserstein Barycenter to the barycenter based on optimal weak mass transport.  Some analytical, statistical and computational properties of this object (the barycenter based on optimal weak mass transport) are discussed and shown. Some numerical experiments are shown too.

**Main Review:**

Apart from some minor typos here and there ( e.g., eq. instead of Eq.) , the paper is well written, and I would say it is tailored for a rather advanced audience that is familiar with the problem of OT and its nuances. With that being said, the introduced concept of weak barycenter is a rather natural extension once once has the weak Wasserstein distance.

I think this paper is very nice as it presents a number of properties for the studied object with some computational an statistical aspects of it. This is the strong part of the paper, and it provides a nice presentation of the problem. However, some of the weaknesses of the present form of  the manuscript need to be improved. Find them below in no particular order.

-  While the weak barycenter is a nice object, I could not find a good or convincing motivation (either computational, statistical, or in applications) for which one would like to compute the weak barycenter.

- In a couple of instances where claims are made with little or no justification, for example in the experiments it is read " weak barycenter being the most 297 concentrated as expected." I have no reason to expect that the weak barycenter will be more concentrated.

- While the relation to the outliers is nice, this comes out of the blue and with no theoretical justification why. Outlier detection is a rather developed field with its own tools to quantify its properties, and I feel this is not developed in this manuscript and read as some experiment to run, with no justification on why robustness is induced. For example, we know that the median is "more robust" than the mean because we need more than 50% of outliers to drive the mean to an arbitrary position. How can we quantify the robustness of the weak barycenter?

- The numerical section is the reviewers opinion the weakest part of the paper. The plots are very nice and interesting, but little to no discussion is made to connect the numerical results to the theoretical properties of the weak barycenter. For example, what is the conclusion from the 8 plots in figure 4 and 5? It is very hard for me to really obtain any new information from them.  Some numerical experiments are needed that really highlight the statistical, computational, and analytical properties of the weak barycenter. I do no see that in the present numerical results.

- Some minimal discussion about the computational complexity of the weak barycenter is needed. Traditional Wasserstein barycenter is know to be very computationally intensive to compute. is the weak barycenter easier or harder to compute?

Minor

-The term "concentrated" is used a couple of times as a property of a probability distribution. Even if this is an intuitive concept, some formal definition would be greatly appreciated.


-

**Time Spent Reviewing:**

4

---

> ### Author Response · Authors · 2021-08-10
> **We justify the potential of the weak barycenter as well as the OWT fot the machine learning community. We also clarify the meaning of "concentrated" and the robustness statement.**
>
> Many thanks for the positive view and encouraging comments. We address the points highlighted in your review as follows.
>
> **(i) Motivation and jutification.** In addition to the specific virtues of the weak barycenter presented at the beginning of our reply to Reviewer eE1U, which justify its use as an alternative to the Wasserstein barycenter, the weak barycenter acts as a Sinkhorn barycenter for mean values of the regularisation parameter (see Fig. 2), and therefore, could be a tool to better understand its behavior. As in the entropy regularisation case, the weak barycenter then seems to present some regularisation properties, possibly linked the robustness (see (iii) below). This line of research is open for future work. Additionally, we believe that optimal weak transport offers promising tools for machine learning challenges, since properties such as the uniqueness of the optimal weak plan are always guaranteed. Hopefully, the weak barycenter method proposed in our work enables the development of related tools for others applications.
>
> **(ii) Meaning of "Concentrated".** We have noted through the experiments that the weak barycenter seems more concentrated than usual Wasserstein barycenter, in the sense that it is less spread out or its samples tend to be closer to each other. This sense of  "concentrated" will be made clearer in the manuscript in terms of the distributions' support.  We acknowledge that the paper did not provide clear theoretical reasons for this phenomenon, so we will rectify our claim about the concentration made regarding one of the experiments.
>
> **(iii) Robustness.** As explained to Reviewer Ee1U (in (ii)), the robustness of the weak barycenter can be related to and stated in terms of its convex order properties. An intuitive and simple way to illustrate this point in the case where the measures $\{\nu_i\}_{i=1,\ldots,n}$ are one dimensional and centered is as follows. By Proposition 2, a weak barycenter $\mu$ must verify $\mu\leq_c\nu_i$ for all $i=1,\ldots,n$. In particular, from Theorem 3.A.1 in [M. Shaked and J.G.Shanthikumar, Stochastic Orders], we get that $\int_x^{\infty} \mathbb{P}[X>u]du \leq \int_x^{\infty} \mathbb{P}[Y_i>u]du$ for all $x\in\mathbb{R}$, where $X\sim \mu$ and $Y_i\sim\nu_i$. Therefore, $\mu$ is likely to avoid outliers.
>
> **(iv) Experiments.** The conclusion of Fig. 4 and 5, as well as for the Pair-of-ellipses experiments (Fig. 9), is that the weak barycenter is more likely to maintain the common (or shared) geometric features of the measures involved, as expected from Theorem 3.
>
> **(v) Computation.** Recently, Jason M. Altschuler, Enric Boix-Adsera published the article "Wasserstein barycenters can be computed in polynomial time in fixed dimension". Until then, Wasserstein barycenters were challenging to compute. However, our proposed weak barycenter is easier to compute than previous techniques as it boils down to compute regularised OT problems. A discussion on the computational complexity will be added.

---

> > ### Comment · Reviewer_qhuc · 2021-08-24
> > **Change my score accordingly**
> >
> > I have read the responses and I believe the main issues have been addressed. I will change my score accordingly.

---

> > > ### Author Response · Authors · 2021-08-28
> > > **Reply**
> > >
> > > Thank you for acknowledging our response. In the new version of the manuscript, we will clarify the meaning of « concentrated » as we have done in this discussion and provide further intuition on the robustness property.

---

### Official Review · Reviewer_Ee1U · 2021-07-17

**Rating:** 6
**Confidence:** 3

**Summary:**

Computing the barycenter of a finite set of probability measures is an important task in various statistical and machine learning problems, and is usually done by comparing the measures with a specific divergence. This divergence plays a significant role as it affects the properties and practical performance of the resulting barycenter, and should then be carefully chosen. Motivated by the attractive theoretical properties of the Wasserstein distance, which emerges from optimal transport (OT) theory, the Wasserstein barycenter has been introduced and extensively studied from a theoretical and practical point a view.

This paper introduces a novel barycenter of measures that is closely related to the Wasserstein barycenter, since the comparisons are made by solving OT problems: the authors introduce the "weak barycenter" (Definition 1), which is based on the recently introduced "optimal weak transport" problem, and study its theoretical properties. Specifically, they prove the existence of a solution to the optimization problem that defines the weak barycenter (Proposition 1), and they provide a characterization of the solutions (Lemma 1, Proposition 2). They also bound the deviation between the weak and Wasserstein barycenters (Lemma 2), and show that the weak barycenter is linked to a specific latent variable (Theorem 3).

Then, the authors introduce the "weak population barycenter" to average an infinite set of probability measures (Definition 2), which can be used for online settings. This barycenter correponds to an optimization problem, which is obtained by extending the definition of the weak barycenter and is well defined (Lemma 3, Proposition 3).

The authors prove that the weak and weak population barycenters are necessarily fixed points of specific operators (Proposition 4, Lemma 4), and they develop two iterative procedures (Algorithms 1 and 2, supplementary document) that are guaranteed to converge to such fixed points (Proposition 5, Theorem 5). The computational requirements of the two algorithms are investigated in Section 6. Finally, several experiments are conducted to compare the empirical performance of the weak barycenter against the Wasserstein barycenter (Section 7).


**Ethical Concerns:**

I don't think this submission raises any ethical issues.

**Limitations And Societal Impact:**

The limitations of this work have been adequately addressed, but not the societal impact.

**Main Review:**

This paper introduces a new methodology to average probability distributions, even if these are provided as a stream. Besides, it is based on optimal transport theory, which has received increased attention in recent years from the statistics, optimization and machine learning community. Therefore, this work is relevant for the NeurIPS community.

The contributions are obtained by adapting existing results from the literature on the Wasserstein barycenter (for example, the two algorithms and their properties are largely inspired by "A fixed-point approach to barycenters in Wasserstein space", Alvarez-Esteban et al.), and in that sense, they are not very original. This does not affect the quality of the work in my opinion: the theoretical findings are technically sound, and I appreciate that the authors provided some explanations on what they imply intuitively (e.g., Lemma 2 and Theorem 3).

However, my main concern, which prevents me from giving a positive score for now, is that it is not clear enough why weak barycenters should be used in practice. Therefore, I strongly encourage the authors to emphasize more the significance of their contribution.
One clear advantage is that the weak population barycenter (and Algorithm 2) can process streaming data. However, results in Section 7.1 and 7.2 only illustrate that the weak population barycenter is more concentrated than the "OT barycenter". On the other hand, the experiment on the cytometry dataset (Section E.3) seems more promising in illustrating the benefits of Algorithm 2, thus could be more discussed in the main document.
Besides, I am not fully convinced by the superior performance of the weak barycenter (and Algorithm 1), since Section 7.3 only shows qualitative results for a single example from the MNIST dataset; the analysis should present more examples supporting this conclusion (e.g., similarly to Section 4 in "Continuous Regularized Wasserstein Barycenters", Li et al.).

While experiments in Section 7 also explore interesting properties that this paper do not study theoretically (the robustness to outliers and relation with OT Sinkhorn barycenters), I think it would be more relevant to illustrate more the consequences of the theoretical results established in Sections 3, for example Lemma 2 and Theorem 3 (the link between robustness to outliers and Theorem 3 is not trivial), as well as the computational aspects of the algorithms (as in Section 5 in "A fixed-point approach to barycenters in Wasserstein space", Alvarez-Esteban et al.).

Regarding clarity, the paper is well written (some minor typos are listed at the end of my review), but quite dense. Besides, I suggest incorporating the following changes in the organization, for better readability:
- A subsection on the Wasserstein barycenter could be added in the background section (Section 2), to recall the definition (l.105-106) and contributions from the literature that are relevant for this work (including Section 6.1).
- Section 3 and 4 could be merged. Same suggestion for Section 5 and 6 (or 6 and 7).
- The explanation on the algorithms for computing the weak barycenters would be much clearer if Algorithms 1 and 2 (or only Algorithm 1) are given in the main document, rather than in the supplementary doc.

Typos:

l.121: missing parentheses for the expectations of $Y_i$

l.230: "an iterative procedure that converge" -> "an iterative procedure that converges"

l.278: "We first focused in Algorithm 2" -> "We first focused on Algorithm 2"

l.303: "$p \in (200, 225)$" -> "$r \in (200, 225)$"


**Time Spent Reviewing:**

8

---

> ### Author Response · Authors · 2021-08-10
> **We highlight the particularities of the weak barycenter, and respond to specific requests regarding the robustness to outliers and computational aspects.**
>
> We thank the reviewer for the positive and thorough comments and for pointing out typos, which will be corrected in the final version.
>
> The main unique property of the weak barycenter is its capability to encode common  geometric information present in all the input measures considered, which can be intuitively and rigorously interpreted as being the distribution of an underlying latent variable. This property is in sharp contrast with the Wasserstein barycenter, which averages the geometric information over all measures. As a consequence, the weak barycenter is particularly robust to the presence of outliers, as explained below in (ii).  The bottom line is that weak barycenter should be preferred on applications requiring the detection of common patterns, e.g., immersed in noise, across multiple datasets, such as our cytometry examples. At a more heuristic level, this feature results in the weak barycenter  being less spread out in the performed experiments, whereas its Wasserstein counterpart is more disperse.
>
> Additionally, we have experimentally shown that the global structure of input datasets is preserved (e.g. see Fig. 9, Pari-of-ellipses), thus further validating the use of the weak barycenter for noise-corrupted observations. Furthermore we believe that optimal weak transport poses general advantages for the machine learning community, and hope that our work will pave the way to various applications. We address the points highlighted in your review as follows.
>
> **(i) Streaming algorithm.** We agree that the streaming algorithm could be further highlighted, especially for applications such as the study of a cytometry dataset. The application of Algorithm 2 to streaming data will be accommodated in the main document.
>
> **(ii) Robustness-to-outliers property of the weak barycenter.** Theorem 3 proves that the weak barycenter can be viewed as the law of a latent variable, whereby  samples of  each of the input measures of the barycenter problem can be generated by adding some "intracluster noise" (and a fixed translation) to it. The observations of each class can thus be interpreted as "outliers" with respect to the (translated) law of the weak barycenter, which are statistically different and are thus left aside of its support.  This way, the weak barycenter is robust to outliers, as it tends to discard them, by construction. Furthermore, this "robustness" property results in the stability of weak barycenter upon perturbation of a class with larger noise (or more scattered, outlying values). More precisely, by Theorem 1  about the weak transport cost and Definition 1 of the weak barycenter, if a class is corrupted in such a way that their observations result in a  stochastically larger distribution than the original one, a weak barycenter computed in terms of the original (stochastically smaller) class will still be a weak barycenter in the new corrupted setting. This explanation will be added in the manuscript in the form of a remark, which we hope will clarify the sense in which the weak barycenter is statistically robust.
>
> **(iii) Computational aspects.** As in Section 5 of "A fixed-point approach to barycenters in Wasserstein space", Alvarez-Esteban et al., we have computed a weak barycenter between two 2D centered ellipses $E(\Sigma_i)=[x\in\mathbb{R}^2\ :\ x^t\Sigma_i^{-1}x = 1]$ with covariance matrices $\Sigma_1 = [2,0;0,1]$ and $\Sigma_2 = [1,0;0,2]$, by considering 300 random observations for each ellipse. We have then executed the iterations of Algorithm 1 until the difference of the objective function (i.e., the sum in Eq.(5)) between two successive iterations was smaller than 1e-5. This occurred at the 8th iteration, for which the obtained weak barycenter was a circle within both ellipses. Moreover, as suggested by Reviewer kNgK, as we have access to the value of the weak barycenter problem (see Eq.(7)), we also compared the value of the objective function at the 8th iteration (that is 3.62e-4) to $\frac{1}{2}\sum_{i=1}^2\Vert \mathbb{E}(Y_i)\Vert^2-\Vert \frac{1}{2} \sum_{i=1}^2 \mathbb{E}(Y_i)\Vert^2$, with a plug-in estimator for $\mathbb{E}(Y_i)$. This way, we obtained an objective of 3.21e-4, therefore, Algorithm 1 gave a satisfying optimised weak barycenter. We will comment on this in the paper.
>
> **(iv) Organisation of the paper.** We will move the definition of the Wasserstein barycenter into the background section. We agree that the length of Sections are uneven, however, we believe that merging Sections of different purposes would not help understanding. We intend to integrate Algorithm 1 in the main document.

---

> > ### Comment · Reviewer_Ee1U · 2021-08-26
> > **Post rebuttal**
> >
> > Thank you for the detailed reply, and for taking the time to conduct an additional experiment. I am now more convinced by the interpretation of Theorem 3, which might indeed explain why weak barycenters seem more concentrated and robust to outliers than traditional OT barycenters. I will increase my score.

---

> > > ### Author Response · Authors · 2021-08-28
> > > **Reply**
> > >
> > > Thank you for you feedback, we will include the clarifications in our response in the manuscript.

---

### Decision · Program_Chairs · 2021-09-27

**Decision:**

Accept (Poster)

**Comment:**

All the reviewers insisted on the quality of the exposition of the paper, which is able to import advanced notions from OT to the ML community. This being said, the paper only partially motivates (from both a theoretical and numerical perspective) the relevance of weak OT for ML. For this reason, and after discussing with the reviewers, I believe that the paper is borderline for acceptance. I nevertheless proposed acceptance. I strongly urge the authors to take into account some of the suggestions of the comments of the reviewers in the final version.